# High-fidelity CRISPR/Cas9- based gene-specific hydroxymethylation rescues gene expression and attenuates renal fibrosis

Xingbo Xu[1,2], Xiaoying Tan[2,3], Björn Tampe [3], Tim Wilhelmi[1,2], Melanie S. Hulshoff[1,2,4], Shoji Saito[3], Tobias Moser [5], Raghu Kalluri[6], Gerd Hasenfuss[1,2], Elisabeth M. Zeisberg[1,2] & Michael Zeisberg[2,3]

While suppression of specific genes through aberrant promoter methylation contributes to different diseases including organ fibrosis, gene-specific reactivation technology is not yet available for therapy. TET enzymes catalyze hydroxymethylation of methylated DNA, reactivating gene expression. We here report generation of a high-fidelity CRISPR/Cas9-based gene-specific dioxygenase by fusing an endonuclease deactivated high-fidelity Cas9 (dHFCas9) to TET3 catalytic domain (TET3CD), targeted to specific genes by guiding RNAs (sgRNA). We demonstrate use of this technology in four different anti-fibrotic genes in different cell types in vitro, among them RASAL1 and Klotho, both hypermethylated in kidney fibrosis. Furthermore, in vivo lentiviral delivery of the Rasal1-targeted fusion protein to interstitial cells and of the Klotho-targeted fusion protein to tubular epithelial cells each results in specific gene reactivation and attenuation of fibrosis, providing gene-specific demethylating technology in a disease model.

[1] Department of Cardiology and Pneumology, University Medical Center Göttingen, Robert-Koch-Str. 40, 37075 Göttingen, Germany. [2] German Center for Cardiovascular Research (DZHK) Partner Site, Göttingen, Germany. [3] Department of Nephrology and Rheumatology, University Medical Center Göttingen, Robert-Koch-Str. 40, 37075 Göttingen, Germany. [4] Department of Pathology and Medical Biology, University Medical Center Groningen, Hanzeplein 1, 9713 Groningen, GZ, Netherlands. [5] Institute for Auditory Neuroscience & Inner Ear Lab, University Medical Center Göttingen, Robert-Koch-Str. 40, 37075 Göttingen, Germany. [6] Department of Cancer Biology, Metastasis Research Center, University of Texas, MD Anderson Cancer Center, 1881 East Road, Houston, TX 77054-1901, USA. These authors contributed equally: Xingbo Xu and Xiaoying Tan. These authors jointly supervised this work: Elisabeth M. Zeisberg and Michael Zeisberg. Correspondence and requests for materials should be addressed to E.M.Z. (email: elisabeth.zeisberg@med.uni-goettingen.de) or to M.Z. (email: mzeisberg@med.uni-goettingen.de)

Aberrant CpG island promoter methylation of select genes leads to silencing of these genes and thus contributes to various pathologies such as cancer, neuronal degeneration, and organ fibrosis[1–3]. Well-studied examples of such genes are RASAL1 (which encodes for a Ras-GAP-like Ras-GTP inhibitor, and hypermethylation of the RASAL1 promoter leads to silencing of RASAL1 expression and increased RAS-GTP activity)[4–6] and KL1 (which encodes for Klotho, a transmembrane protein working as a co-receptor for fibroblast growth factor-23), both of which have been associated with cancer and also fibrogenesis:[7–11] The RASAL1 promoter is consistently hypermethylated in tissue fibrosis including kidney, heart, and liver and also in gastrointestinal cancers[4–6,9,13]. The extent of RASAL1 promoter methylation correlates with progression of kidney fibrosis in patients and mice[14], and rescue of RASAL1 transcription through transgenic overexpression attenuates progression of experimental fibrosis in the kidney[14]. This suggests that reversal of aberrant RASAL1 methylation and rescue of RASAL1 expression are new therapeutic targets to inhibit progression of kidney fibrosis. RASAL1 was originally identified as one of three genes (including also EYA1 encoding for a member of the eyes absent (EYA) family of proteins, which plays a key role in the kidney development [15,16] and LRFN2, encoding for leucine-rich repeat and fibronectin type III domain-containing protein, which functions in presynaptic differentiation[17]) in a genome-wide methylation screen comparing normal and fibrotic kidney fibroblasts, which were consistently downregulated and hypermethylated in fibrotic but not healthy fibroblasts both in humans and mouse[4].

Hypermethylation of the KLOTHO promoter has been shown to be associated with progression of various forms of cancer and to correlate with kidney fibrosis in both humans and experimental fibrosis mouse models[18–22]. In the kidney, Klotho is predominantly expressed in tubular epithelial cells. Reversal of hypermethylated Klotho promoter associated Klotho suppression by a lipophilic anthraquinone compound, Rhein, has been demonstrated to ameliorate renal fibrosis in unilateral ureter obstruction (UUO)-induced fibrotic kidney mouse model. This results through effectively reducing aberrant DNMT1/3a expression and thereby maintaining secreted and membrane Klotho levels[22].

It has long been known that DNA methylation can be inhibited through administration of nucleotide analogs such as 5′azacytidine, which is incorporated into the DNA thereby causing DNA damage and subsequent DNA repair by replacement with unmethylated DNA. While nucleotide analogs are in clinical use in several malignant diseases such as myelodysplastic syndrome as demethylating therapies, they are highly unspecific and their utility is limited to second line therapies due to side effects, highlighting the need for gene-specific, less toxic demethylating therapies.

In this regard, members of the ten-eleven translocation (TET) family of zinc finger proteins (ZFPs) catalyze oxidation of methylated cytosine residues (so-called hydroxymethylation), which subsequently leads to replacement of methylated cytosine residues with naked cytosine[23]. Both hydroxymethylated and demethylated promoters result in re-expression of genes that had been silenced through CpG promoter methylation. We previously demonstrated that (1) TET3 is the predominant TET protein in the kidney[5], (2) kidney fibrosis is associated with decreased TET3 expression[5], and (3) induction of endogenous TET3 expression leads to hydroxymethylation, demethylation, and thereby reactivation of various genes, including RASAL1, within diseased kidneys and attenuates experimental kidney fibrosis[5,14]. TET3 only induces transcription of genes that had been previously methylated, and it is recruited to select genes (including RASAL1) through recognition of a common CXXC motif in proximity to

gene promoter CpG islands, providing enhanced specificity as compared to nucleotide analogs. As opposed to silencing of DNMTs, activation of TET enzymes is an active way of reducing aberrant gene methylation. However, there are more than 9000 genes targeted by TET proteins within the human genome, suggesting gene-specific delivery of TET as an attractive approach to rescue expression of aberrantly methylated genes[24].

Previous studies demonstrated that by fusion of the TET methylcytosine dioxygenase catalytic domain (in which the CXXC binding domain is lacking) to the programmable DNA-binding domains of ZFPs or transcription activator-like effectors (TALE), enhanced gene-specificity of hydroxymethylation and re-expression of methylated genes could be achieved as compared to globally increased TET expression[25,26]. However, utility of these approaches was limited due to off-target effects, high labor intensity, and lack of evidence for disease modifying activities in vivo, revealing that a technique with further enhanced specificity was needed.

Here we aimed to utilize both the high target specificity of sgRNA-guided Streptococcus pyogenes dCas9 and the enzymatic effectiveness of TET3. We demonstrate gene-specific targeting and successful re-expression of hypermethylated genes RASAL1, EYA1, LRFN2, and KLOTHO through all-in-one constructs in which either dCas9 or high-fidelity dCas9, respectively, is fused to the TET3 catalytic domain which is specifically targeted to the promoters of these genes by single-guide RNA (sgRNA). We further systematically established viral targeting of different cell populations in the kidney in vivo and demonstrate that by expression of dCas9/dHFCas9-TET3CD-RASAL1-sgRNA in kidney fibroblasts and of dHFCas9-TET3CD-KLOTHO-sgRNA in epithelial cells, fibrosis is significantly attenuated in a mouse model of kidney fibrosis. In summary, we show that CRISPR/Cas9-based gene-specific hydroxymethylation can rescue gene expression. This technology therefore has a broad application spectrum and may be useful to combat other diseases induced by aberrant gene methylation, such as various forms of cancer and neurodegenerative diseases.

## Results

**Targeted hydroxymethylation rescues gene expression in vitro.** In order to generate a gene-specific hydroxymethylation system, we created a chimeric hydroxymethylase by fusing the TET3 catalytic domain (TET3CD)[24,27] to the C-terminal domain of a double mutated Cas9 (dCas9), in which endonuclease catalytic residues D10A and H840A have been mutated to avoid cutting of the DNA (Figs. 1a, b, Supplementary Fig. 1)[28–33]. We next introduced TET3CD (aa851–aa1795) to generate a hydroxymethylation vector (pLenti-dCas9-TET3CD) (Supplementary Figs. 1, 2)[24,34]. We have also generated a control vector (pLenti-dCas9-TETCDi) in which catalytic residue mutations H1077Y and D1079A have been created to abolish the hydroxymethylation activity of TET3CD (Supplementary Fig. 2)[35]. As proof-of-principle, we aimed to reactivate the genes RASAL1, EYA1, LRFN2, and KLOTHO whose expressions are silenced due to promoter hypermethylation in fibrotic human renal fibroblasts and human tubular epithelial cells, respectively[4,36,37]. To identify applicable sgRNA to enable specific targeting of the dCas9-TET3CD fusion protein to the gene promoters, we designed 10 sgRNAs (five guiding RNAs targeting each strand) targeting the RASAL1 promoter, six sgRNAs (three guiding RNAs targeting each strand) targeting the EYA1 promoter, eight sgRNAs (four guiding RNAs targeting each strand) targeting the LRFN2 promoter and the Klotho promoter, respectively. Those sgRNAs were inserted into the pLenti-dCas9-TET3CD vector to generate gene-specific demethylation constructs for

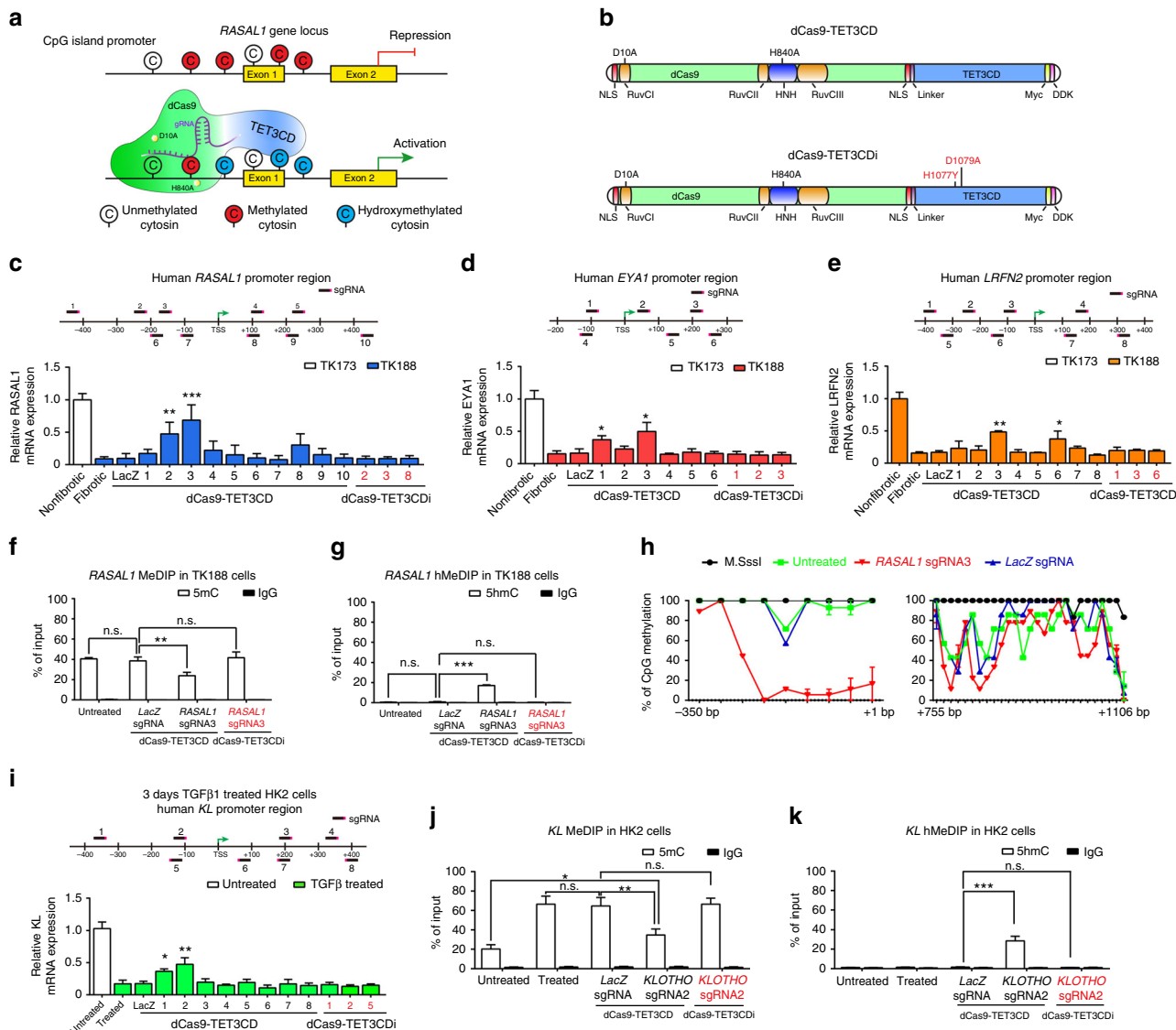

**Fig. 1** Targeted hydroxymethylation of four different aberrantly methylated genes by dCas9-TET3CD fusion protein in human kidney cells. **a** Schematic representing hypermethylated *RASAL1* promoter region (upper panel) and reactivated RASAL1 expression through induction of *RASAL1* promoter hydroxymethylation by dCas9-TET3CD fusion protein in complex with a sgRNA binding to its target region (lower panel). **b** Schematic of domain structure of the dCas9-TET3CD (upper panel) and dCas9-TETCDi (lower panel) fusion protein. **c–e** Locations for *RASAL1/EYA1/LRFN2*-sgRNAs are indicated by thick lines with corresponding PAM in magenta within the human *RASAL1/EYA1/LRFN2* gene locus, respectively. Human fibrotic TK188 fibroblasts were transduced with lentivirus expressing demethylation constructs guided by *RASAL1*-sgRNAs 1–10, *EYA1*-sgRNA 1–6, *LRFN2*-sgRNA 1–8, or by *LacZ* control sgRNA. Results were normalized to reference gene GAPDH. **f, g** MeDIP and hMeDIP analysis of TK188 cells were transduced with dCas9-TET3CD-*RASAL1*-sgRNA3. The results were calculated relative to the input. **h** Bisulfite sequencing summary of promoter methylation status of the *RASAL1* gene in TK188 cells transduced with demethylation constructs guided by *RASAL1*-sgRNA3, by *LacZ* control sgRNA or DNA treated with M.SssI serving as positive control. Each data point represents the mean of three independent transduction experiments with error bars indicating the standard error of the mean for six or more bisulfite sequencing results. **i** Locations for *KL*-sgRNAs are indicated by thick lines with corresponding PAM in magenta within the human *KL* gene locus. Three days TGFβ1-treated HK2 cells were transduced with lentivirus expressing demethylation constructs guided by *KL*-sgRNAs 1–8 or by *LacZ* control sgRNA. **j, k** MeDIP and hMeDIP analysis of HK2 cells were transduction with dCas9-TET3CD-*KL*-sgRNA2. The results were calculated relative to the input. All data are presented as mean value; error bars represent S.D.; $n = 3$ independent biological replicates, n.s. not significant; *$p < 0.05$, **$p < 0.01$, ***$p < 0.001$

*RASAL1* (pLenti-dCas9-*RASAL1*-sgRNA1-10), *EYA1* (pLenti-dCas9-*EYA1*-sgRNA1-6), *LRFN2* (pLenti-dCas9-*LRFN2*-sgRNA1-8), and *KLOTHO* (pLenti-dCas9-TET3CD-KL-sgRNA1-8). *LacZ* sgRNA[38] was introduced into pLenti-dCas9-TET3CD (pLenti-dCas9-*LacZ*-sgRNA) vector serving as control construct.

Upon establishing the demethylation constructs, we tested their demethylation activities utilizing TK188 fibrotic human renal fibroblasts with known robust CpG island promoter methylation

for RASAL1, EYA1, and LRFN2 and HK2 epithelial cells with known CpG island promoter methylation for Klotho upon stimulation with TGFβ1. Fibrotic fibroblasts, which were treated with dCas9-TET3CD-*RASAL1*-sgRNA2/3, showed significant reactivated *RASAL1* expression (Fig.1c). Cells that were treated with dCas9-TET3CD-*EYA1*-sgRNA1/3 showed reactivated *EYA1* expression (Fig. 1d). Cells that were treated with dCas9-TET3CD-*LRFN21*-sgRNA3/6 showed restored LRFN2 expression (Fig. 1e).

Restored expression does not occur with *LacZ* nor with the respective pLenti-dCas9-TETCDi vectors, in which hydroxymethylation activity of TET3CD was abolished (Fig. 1c–e). HK2 epithelial cells showed reduced expression of Klotho upon TGFβ1 treatment, which was restored upon pLenti-dCas9-TET3CD-*KL*-sgRNA1-2 vectors (Fig. 1i). To rule out the possibility that demethylation is due to overexpression of TET3CD, we performed experiments only using TET3CD vectors without guiding RNA. No significant gene induction was observed in any of these 4 genes indicating that demethylation is not due to overexpression of TET3CD (Supplementary Fig. 3). Among the three tested genes in fibroblasts, cells that were treated with dCas9-TET3CD-*RASAL1*-sgRNA3 showed the highest induction

level which was comparable to RASAL1 mRNA expression in control (non-fibrotic) human kidney fibroblasts.

We hence performed methylation- and hydroxymethylation-specific MeDIP and hMeDIP assays (immunoprecipitation of methylated or hydroxymethylated DNA, respectively, followed by qPCR) of the *RASAL1* promoter for dCas9-TET3CD guided by *RASAL1*-sgRNA3, *LacZ*-sgRNA, and dCas9-TET3CDi guided by *RASAL1*-sgRNA3, revealing that among those different vectors, only dCas9-TET3CD-*RASAL1*-sgRNA3 significantly induced *RASAL1* promoter hydroxymethylation and reduced methylation (Fig. 1f, g). To determine if and which CpG sites could be demethylated in the *RASAL1* promoter region after expression of dCas9-TET3CD-*RASAL1*-sgRNA3, bisulfite sequencing was

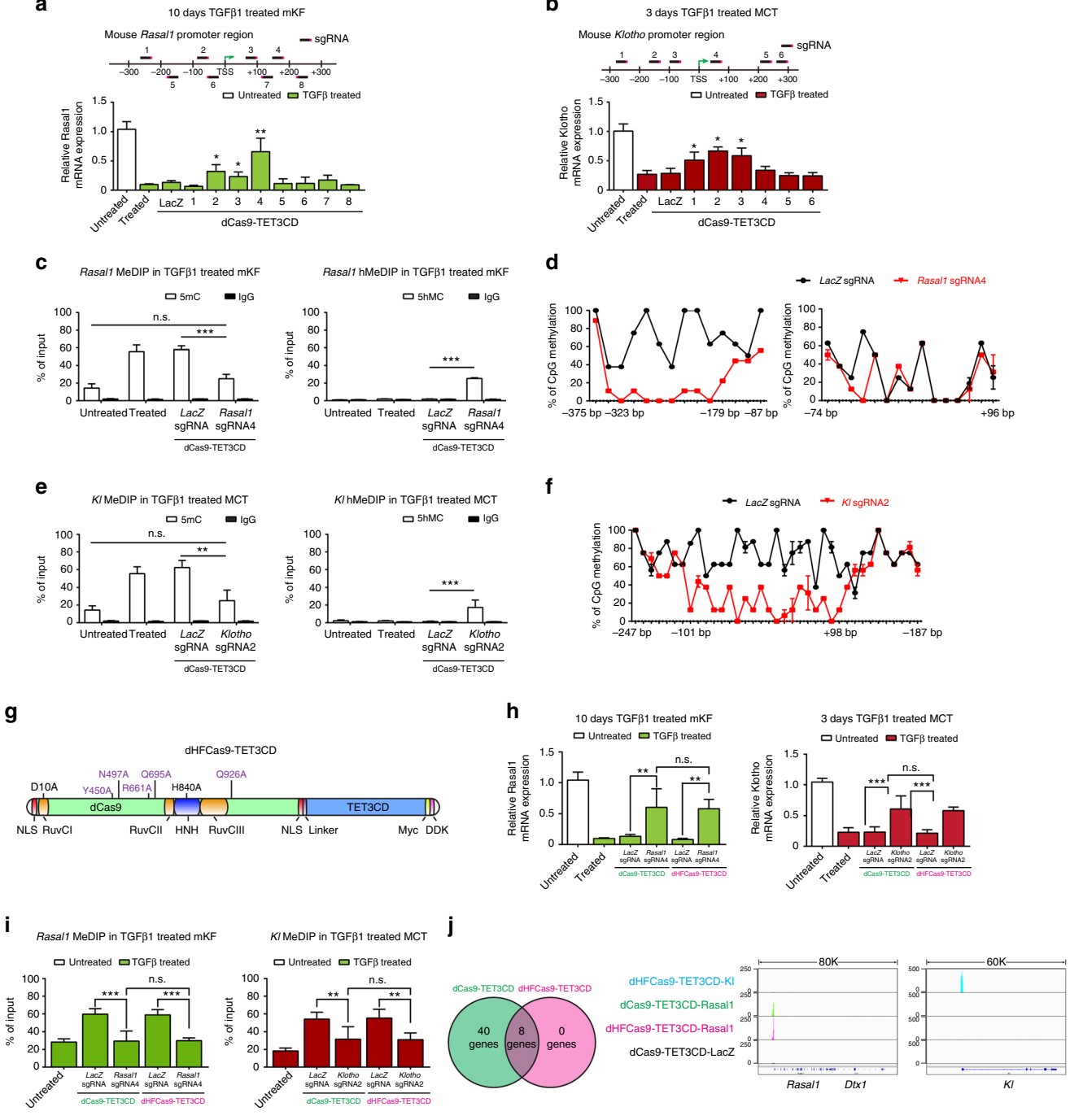

performed. In contrast to dCas9-TET3CD-*LacZ*-sgRNA transduced cells, cells transduced with dCas9-TET3CD-*RASAL1*-sgRNA3 demonstrated demethylation in the promoter region between −114 to +1 (Fig. 1h, Supplementary Fig. 4), suggesting this to be a "critical promoter region". To gain single base-pair resolution for hydroxymethylation within the critical region, we performed glucosylation-mediated restriction enzyme sensitive PCR (gRES-PCR) revealing that upon treatment with T4-BGT and MspI the uncleaved *RASAL1* PCR amplicon was only detectable in the cells transduced with dCas9-TET3CD-*RASAL1*-sgRNA3 (Supplementary Fig. 5d) but not in the other samples (Supplementary Fig. 5b, c, e), confirming site-specific hydroxymethylation of the *RASAL1* promoter by dCas9-TET3CD-*RASAL1*-sgRNA3. Upon establishing that dCas9-TET3CD-*RASAL1*-sgRNA3 effectively induced *RASAL1* CpG promoter hydroxymethylation and demethylation and subsequent rescue of RASAL1 expression, we identified all genes which were predicted to be targeted by sgRNA3 via the online program CCTop[39]. We hence performed qRT-PCR for these genes, but no significant difference in mRNA expression could be detected for any of the predicted genes other than RASAL1 when comparing cells transduced with dCas9-TET3CD-*RASAL1*-sgRNA3 with cells transduced with dCas9-TET3CD-*LacZ*-sgRNA (Supplementary Fig. 6, Supplementary Table 8). This suggests that none of these predicted off-targeted genes was hypermethylated and hence they were not affected by our demethylation system.

Just as for RASAL1, we rescued KL expression corresponding with enhanced *KL* promoter hydroxymethylation and decreased promoter methylation (Fig. 1j, k). We also performed qRT-PCR for those genes that were predicted to be *KL* sgRNA2 off-targets. Other than KL no significant difference in mRNA expression could be detected when comparing cells transduced with sgRNA2 with cells transduced with *LacZ*-sgRNA (Supplementary Fig. 7, Supplementary Table 9).

In summary, we demonstrate successful targeted demethylation of 4 different aberrantly methylated gene promoters (*RASAL1*, *EYA1*, *LRFN1*, and *KL*) and in two different cell types (fibroblasts and epithelial cells) through lentiviral delivery of a construct encoding a fusion protein of dCas9-TET3CD, which is targeted to the promoter CpG through specific single-guide RNA. Notably, we realized that demethylation proteins guided by sgRNAs that are targeting the DNA antisense strand are more efficient as compared to those where the DNA sense strand was targeted.

In order to explore the possibility of utilizing this demethylation system in mice in vivo, we next tested our system in primary mouse kidney fibroblasts (mKFs) and in mouse renal tubular epithelial cells (MCT cells), where Rasal1 and Kl expressions are reduced, respectively, via promoter hypermethylation by prolonged exposure to TGFβ1 treatment (Fig. 2a–f, Supplementary Fig. 8a, b)[4,5]. We designed eight different sgRNAs (four sgRNAs targeting each strand) targeting the *Rasal1* promoter and six different sgRNAs (all sgRNAs targeting antisense strand) targeting the *Kl* promoter, and introduced them into a pLenti-dCas9-TET3CD vector (pLenti-dCas9-TET3CD-*Rasal1*-sgRNA1-8 or pLenti-dCas9-TET3CD-*Kl*-sgRNA1-6) to transduce 10 days TGFβ1-treated mKFs or three days TGFβ1-treated MCT cells, respectively. We identified that three of the constructs (dCas9-TET3CD-*Rasal1*-sgRNA2-4) rescued Rasal1 expression (Fig. 2a, Supplementary Fig. 8d) and three of the constructs (dCas9-TET3CD-*Kl*-sgRNA1-3) restored *Kl* expression (Fig. 2b). Moreover, rescued Rasal1 and Kl expression corresponded with enhanced hydroxymethylation and attenuated promoter methylation for both genes (Fig. 2a–f). Bisulfite sequencing identified −323 bp to −179 bp as the critical region within the murine *Rasal1* promoter which has been demethylated (Fig. 2d, Supplementary Fig. 9) and −101 bp to +98 bp as the critical region within the murine *Kl* promoter which has been effectively demethylated (Fig. 2f, Supplementary Fig. 10).

In order to test off-target effects (a problem immanent to the Cas9 technology, which has thus far limited therapeutic utility[40]), we performed chromatin immunoprecipitation followed by sequencing (ChIP-seq) for dCas9-TET3CD-*Rasal1*-sgRNA4 binding sites in mKFs. Our data reveal the targeted region of the *Rasal1* promoter and a large number of 159 off-target binding sites within 48 different genes (Table 1). Importantly, these off-target genes included genes with known pro-fibrotic effects (which are commonly silenced by promoter methylation in normal tissue) such that therapeutic efficacy by rescue of (anti-fibrotic) *RASAl1* could be counteracted by newly induced expression of pro-fibrotic genes through our dCas9-TET3CD construct. We therefore next aimed to improve our technique to reduce off-target effects. Among all methods to reduce off-target effects of Cas9, the use of high-fidelity CRISPR-spCas9 has been shown to be the most efficient[41]. We hence introduced catalytic domain deactivation amino acid mutations (D10A and H840A) into high-fidelity spCas9 to abolish its cleavage properties and we

**Fig. 2** dCas9-TET3CD and dHFCas9-TET3CD fusion proteins induce targeted *Rasal1/ Kl* promoter demethylation in mouse kidney cells and dHFCas9-TET3CD largely reduced off-target effects. **a** Locations for *Rasal1*-sgRNAs are indicated by thick lines with corresponding PAM in magenta within the mouse *Rasal1* gene locus. 10 days TGFβ1-treated mKF were transduced with dCas9-TET3CD-*Rasal1*-sgRNAs1-8 or by *LacZ* control sgRNA. **b** Locations for *Klotho*-sgRNAs are indicated by thick lines with corresponding PAM in magenta within the mouse *Klotho* gene locus. Three days TGFβ1-treated MCT were transduced with dCas9-TET3CD-*Klotho*-sgRNAs1-6 or by *LacZ* control sgRNA. **c** MeDIP and hMeDIP analysis of TGFβ1-treated mKF were transduced with dCas9-TET3CD-*Rasal1*-sgRNA4. **d** Bisulfite sequencing summary of promoter methylation status of the *Rasal1* gene in TGFβ1-treated cells transduced with dCas9-TET3CD-*Rasal1*-sgRNA4 or by *LacZ* control sgRNA. **e** MeDIP and hMeDIP analysis of TGFβ1-treated MCT cells transduced with dCas9-TET3CD-*Kl*-sgRNA2 or with *LacZ* control sgRNA. **f** Bisulfite sequencing summary of promoter methylation status of the *Klotho* gene in TGFβ1-treated MCT cells transduced with dCas9-TET3CD-sgRNA2 or by *LacZ* control sgRNA. **g** Schematic of domain structure of the dHFCas9-TET3CD fusion protein. **h** TGFβ1-treated mKFs were transduced with dCas9/dHFCas9-TET3CD-*Rasal1*-sgRNA4 or *LacZ* control sgRNA (left panel). TGFβ1-treated MCT cells were transduced with dCas9/dHFCas9-TET3CD-*Klotho*-sgRNA2 or *LacZ* control sgRNA (right panel). **i** MeDIP-qPCR analysis of TGFβ1-treated mKFs transduced with dCas9/dHFCas9-TET3CD-*Rasal1*-sgRNA4 or *LacZ* control sgRNA (left panel) and TGFβ1-treated MCT cells transduced with dCas9/dHFCas9-TET3CD-*Klotho*-sgRNA2 or *LacZ* control sgRNA (right panel). **j** Venn diagram summarizes the common off-targets identified by ChIP-seq analysis between dCas9-TET3CD and dHFCas9-TET3CD transduced mKF (left panel). Tracks indicate the binding regions and the enrichment of dCas9/dHFCas9-TET3CD-*Rasal1/Klotho/LacZ* sgRNA protein-RNA complexes in mKF cells as visualized in the IGV browser (right panel). Genomic coordinates are shown below the tracks (build mm9). All data are presented as mean value; error bars represent S.D, $n = 3$ independent biological replicates, n.s. not significant, *$p < 0.05$, **$p < 0.01$, ***$p < 0.001$. qRT-PCR results were normalized to reference gene Gapdh. MeDIP and hMeDIP results were calculated relative to input. For bisulfate sequencing each data point represents the mean of three independent biological replicates with error bars indicating the standard error of the mean for six or more bisulfite sequencing results

**Table 1 Off-target genes identified by ChIP-sequencing analysis**

**Targets of dCas9-TET3CD-*Rasal1*-sgRNA4**

|   | Chrom | Start | End | Score | ThickStart | ThickEnd | ItemRGB | BlockCount | Gene symbol |
|---|---|---|---|---|---|---|---|---|---|
| 1 | chr5 | 114153013 | 114153177 | 65 | 6.52912 | 11.72342 | 6.56068 | 89 | 1700069L16Rik |
| 2 | chr12 | 112968908 | 112969072 | 47 | 5.59639 | 9.73582 | 4.71895 | 68 | 2810002N01Rik |
| 3 | chr16 | 57391581 | 57391772 | 173 | 9.89721 | 23.05536 | 17.33657 | 48 | 2610528E23Rik |
| 4 | chr6 | 67692952 | 67693120 | 140 | 10.66121 | 19.54378 | 14.08762 | 37 | abParts |
| 5 | chr8 | 125333034 | 125333227 | 47 | 5.59639 | 9.73582 | 4.71895 | 96 | Acsf3 |
| 6 | chr2 | 33906296 | 33906460 | 65 | 6.52912 | 11.72342 | 6.56068 | 112 | AK048710 |
| 7 | chr6 | 4764451 | 4764703 | 65 | 6.52912 | 11.72342 | 6.56068 | 92 | AK076963 |
| 8 | chr7 | 69254460 | 69254632 | 180 | 12.43808 | 23.6066 | 18.03127 | 30 | AK086712 |
| 9 | chr16 | 50557448 | 50557617 | 30 | 4.66366 | 7.81498 | 3.03828 | 84 | AK144624 |
| 10 | chr5 | 4053134 | 4053342 | 47 | 5.59639 | 9.73582 | 4.71895 | 104 | Akap9 |
| 11 | chr3 | 126852658 | 126852848 | 59 | 6.87729 | 10.96568 | 5.9056 | 56 | Ank2 |
| 12 | chr6 | 86698604 | 86698768 | 47 | 5.59639 | 9.73582 | 4.71895 | 82 | Anxa4 |
| 13 | chr1 | 50872914 | 50873114 | 465 | 23.98771 | 52.41908 | 46.51371 | 49 | BC029710 |
| 14 | chr17 | 13745750 | 13745921 | 114 | 4.18092 | 16.85615 | 11.47872 | 45 | BC068229 |
| 15 | chr10 | 58926851 | 58927510 | 247 | 36.82354 | 254.31743 | 247.07439 | 36 | Ccdc109a |
| 16 | chr10 | 94156808 | 94156973 | 105 | 8.39458 | 15.86543 | 10.50555 | 90 | Ccdc41 |
| 17 | chr11 | 51843853 | 51844017 | 30 | 4.66366 | 7.81498 | 3.03828 | 55 | Cdk13 |
| 18 | chr11 | 12213679 | 12213882 | 65 | 6.52912 | 11.72342 | 6.56068 | 45 | Cob1 |
| 19 | chr7 | 26797237 | 26797401 | 76 | 7.92883 | 12.87339 | 7.67306 | 27 | Cyp2b6 |
| 20 | chr3 | 102939715 | 102939879 | 47 | 5.59639 | 9.73582 | 4.71895 | 114 | Dennd2c |
| 21 | chr5 | 121098768 | 121099300 | 840 | 1.56069 | 13.73752 | 8.49179 | 341 | Rasal1 |
| 22 | chr14 | 8703442 | 8703750 | 40 | 6.77979 | 9.17137 | 4.05255 | 190 | Flnb |
| 23 | chr7 | 34880923 | 34881149 | 150 | 11.29756 | 20.53988 | 15.0373 | 140 | Gm12776 |
| 24 | chr6 | 142752466 | 142752630 | 47 | 5.59639 | 9.73582 | 4.71895 | 76 | Gm766 |
| 25 | chr8 | 96357751 | 96357949 | 105 | 8.39458 | 15.86543 | 10.50555 | 95 | Gnao1 |
| 26 | chr5 | 104423124 | 104423288 | 65 | 6.52912 | 11.72342 | 6.56068 | 70 | Hsd17b11 |
| 27 | chr13 | 13846763 | 13846961 | 65 | 6.52912 | 11.72342 | 6.56068 | 82 | Lyst |
| 28 | chr2 | 4930750 | 4931328 | 282 | 15.85643 | 33.99368 | 28.26957 | 289 | Mcm10 |
| 29 | chr18 | 7609200 | 7609768 | 168 | 11.19277 | 22.41256 | 16.85413 | 149 | Mpp7 |
| 30 | chr13 | 100159145 | 100159461 | 84 | 7.46185 | 13.7689 | 8.49179 | 232 | Mrps27 |
| 31 | chr10 | 6271609 | 6271773 | 84 | 7.46185 | 13.7689 | 8.49179 | 76 | Mthfd1l |
| 32 | chr14 | 58173518 | 58173682 | 84 | 7.46185 | 13.7689 | 8.49179 | 94 | N6amt2 |
| 33 | chr15 | 95099096 | 95099260 | 65 | 6.52912 | 11.72342 | 6.56068 | 142 | Nell2 |
| 34 | chr9 | 121151 | 121315 | 47 | 5.59639 | 9.73582 | 4.71895 | 111 | Nlrp4g |
| 35 | chr7 | 24212257 | 24212603 | 213 | 13.05824 | 26.95684 | 21.33005 | 163 | Nlrp5 |
| 36 | chr12 | 83086950 | 83087140 | 47 | 5.59639 | 9.73582 | 4.71895 | 35 | Pcnx |
| 37 | chr3 | 96646134 | 96646506 | 65 | 6.52912 | 11.72342 | 6.56068 | 181 | Pdzk1 |
| 38 | chr3 | 152449257 | 152449918 | 190 | 12.12551 | 24.66862 | 19.07566 | 340 | Pigk |
| 39 | chr9 | 66725484 | 66725736 | 84 | 7.9959 | 13.71865 | 8.4867 | 126 | Rab8b |
| 40 | chr4 | 120536872 | 120537036 | 84 | 7.46185 | 13.7689 | 8.49179 | 82 | Rims3 |
| 41 | chr17 | 8149373 | 8149587 | 47 | 5.59639 | 9.73582 | 4.71895 | 57 | Rsph3a |
| 42 | chr4 | 112068812 | 112069093 | 213 | 13.05824 | 26.95684 | 21.33005 | 143 | Skint9 |
| 43 | chr6 | 142181926 | 142182108 | 378 | 19.58736 | 43.71904 | 37.89215 | 36 | Slc21a7 |
| 44 | chr1 | 57911770 | 57911934 | 47 | 5.59639 | 9.73582 | 4.71895 | 35 | Spats21 |
| 45 | chr10 | 5264302 | 5264480 | 47 | 6.52912 | 11.72342 | 6.56068 | 99 | Syne1 |
| 46 | chr5 | 64639102 | 64639266 | 65 | 6.52912 | 11.72342 | 6.56068 | 66 | Tbc1d1 |
| 47 | chr4 | 74361648 | 74361927 | 187 | 13.99224 | 24.4904 | 18.76297 | 145 | Tmem56 |
| 48 | chr1 | 43834505 | 43834730 | 65 | 6.52912 | 11.72342 | 6.56068 | 124 | Uxs1 |

**Targets of dHFCas9-TET3CD-*Rasal1*-sgRNA4**

|   | Chrom | Start | End | Score | ThickStart | ThickEnd | ItemRGB | BlockCount | Gene symbol |
|---|---|---|---|---|---|---|---|---|---|
| 1 | chr1 | 50872919 | 50873111 | 322 | 19.47483 | 38.10745 | 32.22037 | 39 | BC029710 |
| 2 | chr17 | 13745609 | 13745912 | 88 | 3.17695 | 14.27044 | 8.84702 | 170 | BC068229 |
| 3 | chr10 | 58926856 | 58927475 | 359 | 23.74614 | 193.00722 | 185.91823 | 82 | ccdc109a |
| 4 | chr14 | 8703110 | 8703279 | 73 | 8.12621 | 12.74668 | 7.36305 | 122 | Flnb |
| 5 | chr7 | 34880926 | 34881147 | 169 | 13.16969 | 22.59302 | 16.90299 | 132 | Gm12776 |
| 6 | chr5 | 121098997 | 121099271 | 1273 | 1.52457 | 12.67083 | 7.3371 | 425 | Rasal1 |
| 7 | chr3 | 120952807 | 120952979 | 152 | 11.73355 | 20.8748 | 15.21403 | 37 | Tmem56 |
| 8 | chr10 | 5264302 | 5264480 | 47 | 5.59639 | 9.73582 | 4.71895 | 57 | Syne1 |

generated a new dHFCas9-TET3CD demethylation construct (Fig. 2g, Supplementary Fig. 11).

We used mouse *Rasal1* sgRNA4 and *Kl* sgRNA2 (which effectively induced gene expression in dCas9-TET3CD constructs) to generate dHFCas9-TET3CD-*Rasal1*-sgRNA4 and dHFCas9-TET3CD-*Kl*-sgRNA2 and transduced them into mKFs or MCT cells, respectively. Both Rasal1 and Klotho expressions were significantly reactivated to the same level as induced by the dCas9-TET3CD vectors (Fig. 2h). Furthermore, rescued Rasal1 and Kl gene expression corresponded with attenuated promoter

methylation to the same level as induced by the dCas9-TET3CD vectors (Fig. 2i). To compare off-target binding sites between dCas9-TET3CD and dHFCas9-TET3CD, we performed ChIP-seq on cells transfected with dCas9-TET3CD-*LacZ*-sgRNA/-*Rasal1*-sgRNA4, or dHFCas9-TET3CD-*LacZ*-sgRNA/-*Rasal1*-sgRNA4. Using this approach, only eight genes were common for both dCas9-TET3CD and dHFCas9-TET3CD (Table 1), besides 40 peaks specific only for dCas9-TET3 where many of the off-target peaks showed quite high binding levels, as defined by the peak height relative to on-target peaks after subtracting dCas9-TET3CD-*LacZ*-sgRNA reads at that site (Fig. 2j, Table 1).

**RASAL1 knockdown aggravates kidney fibrosis in vivo**. To perform proof-of principle experiments for a therapeutic efficacy of Rasal1 demethylation constructs, we next generated a *Rasal1* knockout mouse model to validate that loss of Rasal1 is pro-fibrotic. It has been demonstrated by different research groups that fibrosis in the kidney, heart and liver, and also gastro-intestinal cancers are associated with *RASAL1* promoter hyper-methylation, and that RASAL1 promoter hypermethylation leads to decreased RASAL1 expression[4–6,9,13]. Also, the extent of *RASAL1* promoter methylation correlates with progression of kidney fibrosis in patients and mice[14], and rescue of RASAL1 transcription through transgenic overexpression attenuates experimental fibrosis in the kidney[14]. However, whether loss of Rasal1 expression per se contributes to kidney fibrosis has not yet been addressed.

We therefore generated *Rasal1*tm1a/tm1a mice that harbor a gene-trap DNA cassette consisting of a splice acceptor site, an internal ribosome entry site and a β-galactosidase reporter, inserted into the second intron of the gene as described extensively in the methods section (Fig. 3a and Supplementary Fig. 12). In these mice Rasal1 expression is reduced by 80% on mRNA and protein level as compared to wild-type littermate control mice (Fig. 3b, c). Kidneys of *Rasal1*tm1a/tm1a mice appeared unchanged as compared to wild-type littermate control mice under baseline condition. However, after challenge with UUO, a model of obstructive nephropathy, which results in severe kidney fibrosis through increased parenchymal pressure, ensuing ischemia, tubular epithelial cell death, and inflammation 7 days after surgery. *Rasal1*tm1a/tm1a mice displayed significantly higher levels of kidney fibrosis as compared to wild-type littermate controls, associated with higher levels of Collagen-1 deposition and abundance of a-SMA-positive fibroblasts (Fig. 3d).

**Viral delivery mode affects cellular targeting in vivo**. After establishing that lack of Rasal1 aggravates kidney fibrosis and in light of known causality of lack of Klotho as another contributor to kidney fibrosis and importantly because both genes are silenced by hypermethylation during kidney fibrosis, we decided to test if targeted hydroxymethlation of Rasal1 and Klotho, respectively, ameliorates kidney fibrosis in vivo. Because Rasal1 is primarily expressed in fibroblasts and Klotho in tubular epithelial cells, we first established if different modes of lentiviral delivery impact what cells are primarily targeted.

We therefore used green and red fluorescence protein (GFP and RFP) labeled control lentivirus to transduce kidneys by different delivery routes. Transduction efficiency was > 95% in mouse kidney fibroblasts and > 95% in mouse kidney epithelial cells in vitro (Fig. 4a, b). Next, we analyzed which cells are targeted in vivo by testing four different delivery methods (using $10^8$ TU/80 μl virus particles each) in both healthy and UUO kidneys: via the renal artery, intraparenchymal (4 sites with 20 μl/site), via the renal vein, and via retrograde infusion into the ureter. Ten days after virus injection, mice were sacrificed and

GFP and RFP expression was visualized by immunohistochemistry using antibodies against GFP and RFP, respectively (Fig. 4c–f). Injection of lentivirus into the renal artery transduced the fewest cells overall among all four techniques in both healthy and UUO kidneys (Fig. 4c). Venous injection leads to transduction of interstitial cells (Fig. 4e), albeit to a lower extent as compared to intraparenchymal injection (Fig. 4d). In contrast, retrograde injection into the ureter predominantly transduced tubular epithelial cells with high efficacy (Fig. 4f).

In order to quantify percentage of transduced fibroblasts we next performed further analysis by double labeling of α-smooth muscle actin (αSMA)-positive fibroblasts and GFP. Ten days after lentiviral intraparenchymal CMV-GFP construct delivery and UUO surgery, an average of 46% of all αSMA-positive fibroblasts were also positive for GFP indicating successful transduction (Fig. 4g).

**Hydroxymethylation of *Rasal1* and *Klotho* improves kidney fibrosis**. To test the efficacy of the established dCas9-TET3CD-*Rasal1*-sgRNA and dCas9-TET3CD-*Kl*-sgRNA systems in vivo, we utilized the mouse model of UUO. This model displays robust *Rasal1* promoter methylation within interstitial fibroblasts and *Klotho* methylation in tubular epithelial cells, resulting in transcriptional suppression of *Rasal1* and *Klotho*, which causes disease progression[4,5]. Based on our previous results which demonstrated effective lentiviral transduction of kidney interstitial cells upon vector delivery through intraparenchymal injection and of epithelial cells upon vector delivery through retrograde ureter infusion, we first injected lentivirus harboring either dCas9-TET3CD-/*Rasal1* sgRNA4 or dHFCas9-TET3CD-/*Rasal1* sgRNA4 with dCas9-TET3CD-*LacZ*-sgRNA or dHFCas9-TET3CD-*LacZ*-sgRNA as controls, respectively, into the renal parenchyme of UUO-challenged and contralateral control kidneys (Fig. 5a, b).

Restored Rasal1 expression upon UUO was observed exclusively in mice which received either dCas9-TET3CD-*Rasal1*-sgRNA4 or dHFCas9-TET3CD-*Rasal1*-sgRNA4, but not in mice injected with the dCas9/dHFCas9-TET3CD-*LacZ*-sgRNA control vectors (Fig. 5c, d). Increased Rasal1 expression correlated with increased *Rasal1* hydroxymethylation and reduced methylation (Fig. 5e, f). Most importantly, renal fibrosis, accumulation of fibroblasts and type I collagen were significantly attenuated in dCas9/dHFCas9-TET3CD-*Rasal1*-sgRNA4 treated mice, but not in mice administered with the dCas9/dHFCas9-TET3CD-*LacZ*-sgRNA control vectors (Fig. 5g–j). Interestingly, even though *Rasal1* hydroxymethylation and restoration of Rasal1 expression was equally effective, attenuation of kidney fibrosis was almost 50% and thereby more effective in dHF- as compared to less than 30% in dCas9-TET3CD-*Rasal1*-sgRNA treated mice (Fig. 5c–j), which is likely due to the reduction of off-target effects in the dHFCas9-TET3CD as compared to the dCas9-TET3CD system (Fig. 2j).

After establishing that dHFCas9-TET3CD is superior to the dCas9-TET3CD system, we continued to use the dHFCas9-TET3CD system for targeted hydroxymethylation of *Klotho* in tubular epithelial cells in order to test its anti-fibrotic potential in vivo. To target methylated *Klotho* in tubular epithelial cells, we performed retrograde injection of dHFCas9-TET3CD-*Klotho*-sgRNA and dHFCas9-TET3CD-*LacZ*-sgRNA control viruses into the ureters of UUO-challenged and of contralateral control kidneys and analyzed the kidneys after 10 days (Fig. 6a, b). Similar as with Rasal1, Klotho expression was successfully restored to ~50% of the physiological level by dHFCas9-TET3CD-*Klotho*s-gRNA but not with *LacZ*-sgRNA and restoration correlated with reduced *Klotho* promoter methylation levels

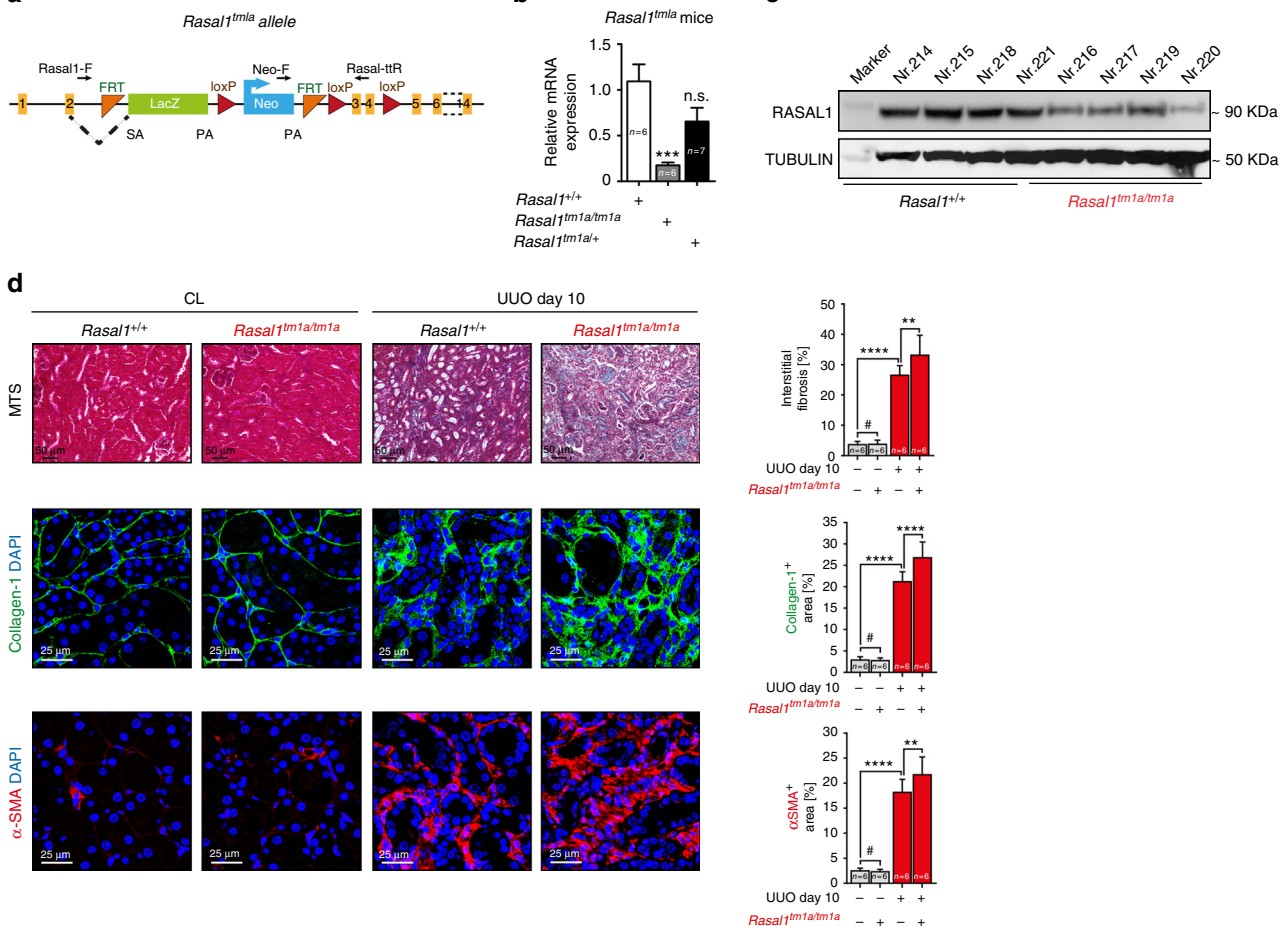

**Fig. 3** *Rasal1* gene disruption results in aggravated fibrosis level in the UUO model. **a** Schematic of knockout-first strategy for *Rasal1* gene. The gene trapping *LacZ* cassette is alternatively spliced with exon 2 mediated by a splicing acceptor (SA). A promoter-driven Neomycin cassette is inserted after *LacZ*. The targeted exon 3 and 4 are flanked by loxP sites. Black arrows indicate the location of genotyping primers. **b** qRT-PCR analysis shows the Rasal1 mRNA expression in homozygous mice are significantly reduced as compared with wild-type mice, while there is no significant reduction in heterozygous mice. The data are presented as mean value, error bars represent S.D., n.s. not significant, ***$p < 0.001$. **c** Western blot analysis shows the RASAL1 protein expression is largely decreased in homozygous mice as compared with wild-type mice. **d** Kidney sections of wild type and homozygous *Rasal1*[tm1a] mutant mouse which were either sham controls (CL) or challenged with UUO were stained for Masson's Trichrome (MTS) (representative light microscopy images are shown in the top row), Collagen-1 or α-SMA (representative confocal images are shown in the middle and bottom row, respectively.) (Scale bars: 25 μm or 50 μm). Quantification of the percentage of total interstitial fibrosis and immunostained positive cells in each group are depicted (data are presented as mean value, error bars represent S.E.M., # not significant, ****$p < 0.0001$)

(Fig. 6c–f). Kidney fibrosis was significantly reduced by 25.4% by dHFCas9-*Klotho*-sgRNA as compared to dHF-TET3CD-*LacZ*-sgRNA injection (Fig. 6f), and this reduction in fibrosis correlated with blunted accumulation of fibroblasts and of type I Collagen (Fig. 6g, h), correlating with *Klotho* demethylation and rescued Klotho expression.

## Discussion

In this study, we provide proof-of-principle that by using a novel dCas9/dHFCas9-TET3CD all-in-one fusion protein approach, single methylated genes can be specifically targeted and transcriptionally reactivated in vitro as well as in vivo in a disease model. Based on the example of four different genes (*RASAL1*, *EYA1*, *LRFN2*, and *KL*) that are known to be hypermethylated in specific cell types or upon stimulation with TGFβ1, we demonstrate that targeted TET3-mediated hydroxymethylation is a feasible, reliable, and fast technology which results in demethylation and transcriptional reactivation of these genes. Because we also demonstrate that use of the mutated wild-type SpCas9 in this technology results in substantial off-target effects we developed a

new high-fidelity Cas9-based approach which reduced off-target genes by 85%. The relevance of reduction of off-target genes was proven by testing our gene-specific demethylation technologies in a disease model in vivo, which to our knowledge has not been done before.

Among the four genes for which we established gene-specific hydroxymethylation vectors, we selected *Rasal1* and *Klotho* for in vivo studies, as both genes have been well studied in context of kidney fibrosis in both human and mouse models: Klotho has been shown to be hypermethylated and transcriptionally silenced in kidney fibrosis patients and in corresponding mouse models, and lack of Klotho is causally linked to kidney fibrosis in mice. Rasal1 has been shown to be transcriptionally silenced and hypermethylated in both human and mouse kidney fibrosis. Because the causality between Rasal1 and kidney fibrosis had not yet been addressed, we generated Rasal1 knockout mice in which Rasal1 expression was reduced by 70%. In these mice, kidney fibrosis was substantially increased upon challenge with UUO, thus causally linking lack of Rasal1 with kidney fibrosis.

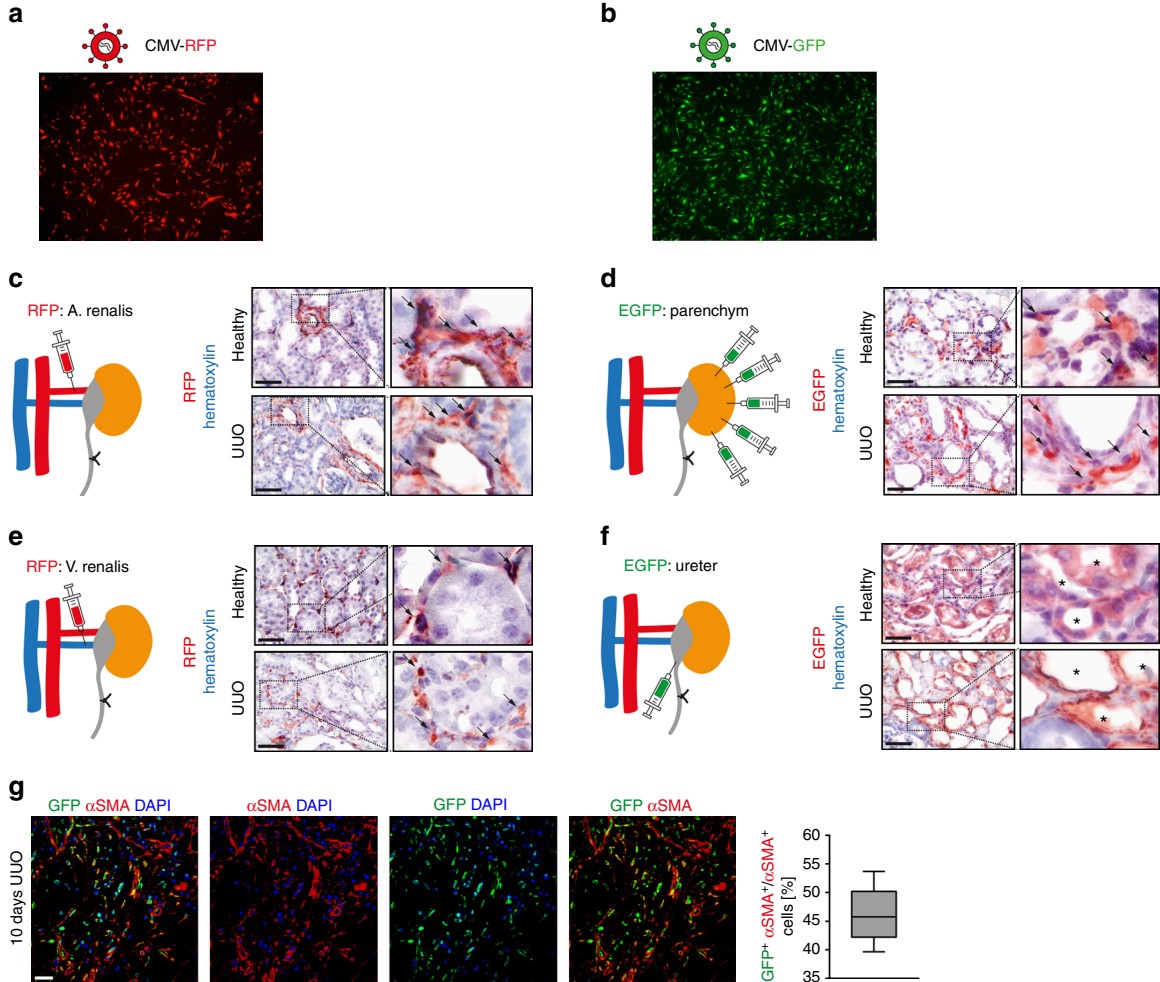

**Fig. 4** In vivo gene delivery to mouse kidney by Lentivirus transduction. **a, b** Immunofluorescence pictures show that mouse kidney fibroblasts and kidney epithelial cells were efficiently transduced with lentivirus containing a RFP (**a**) or EGFP (**b**) gene, respectively. **c–f** Immunohistochemistry pictures show RFP-/EGFP-positive cells in kidney sections which were transduced with lentivirus containing RFP (**c**, **e**) or EGFP (**d**, **f**) through different delivery routes: renal artery (**c**), parenchyma (**d**), renal vein (**e**), and infusion into retrograde ureter (**f**) with 80 μl ($10^8$ TU) virus particles for each method. $n = 6$ in each injection group. **g** Representative confocal photomicrographs of UUO kidneys transduced with GFP-labeled lentivirus via parenchymal injection. The sections were double stained for GFP (in green) and myofibroblast marker α-SMA (in red). Nuclei were counterstained with DAPI (in blue). The box-whisker plot (right panel) shows the percentage of GFP and α-SMA double positive cells out of all α-SMA-positive cells. $n = 3$ mice

Predominant expression of Rasal1 occurs in kidney fibroblasts and of Klotho in tubular epithelial cells. Both cell types are separated by a basal membrane. Because it has been shown that lentiviral constructs do not cross basal membranes[42], we established different routes of lentiviral delivery to primarily target interstitial cells (via parenchymal injection) or epithelial cells (via the ureter). By these respective modes of injection we were able to specifically reactivate Klotho expression in tubular epithelial cells by dHFCas9-TET3CD-Klotho-sgRNA and Rasal1 expression in interstitial cells by dCas/dHFCas9-TET3CD-Rasal1-sgRNA constructs in the UUO mouse model of kidney fibrosis and to ameliorate kidney fibrosis. Interestingly, the therapeutic anti-fibrotic effect of dCas9-TET3CD-Rasal1-sgRNA construct was much smaller (less than 30% fibrosis reduction) as compared to the dHFCas9-TET3CD-Rasal1-sgRNA (almost 50% reduction in total interstitial fibrosis) despite a complete reactivation of Rasal1 expression by both constructs. It appears likely that this is due to off-target effects of dCas9-TET3CD which reactivated pro-fibrotic genes *Anxa4*, and *Nlrp5* (off-targeted by dCas9-TET3CD but not by dHFCas9-TET3CD, Table 1) along with

*Rasal1*, thus limiting the therapeutic effect and highlighting the need for the use of high-fidelity Cas9 in this context.

In general, by fusing dCas9 with the TET3 catalytic domain, we achieved superior specificity and reached a more extended region of demethylation from the target site as compared to previous Zinc finger and TALE-based approaches. Previous studies showed that TET3 mediates hydroxymethylation of cytosine bases (e.g., within the *Rasal1* promoter) followed by consecutive Tdg-mediated base excision, and subsequent replacement with naked cytosine[14]. siRNA-mediated depletion of Tdg effectively prevented *Rasal1* demethylation, suggesting that Tdg is needed for the return to baseline gene expression upon hydroxymethylation[14]. Our study is in line with previous reports which demonstrated gene-specific reactivation of epigenetically silenced genes within cultured cells using dCas9-p300, dCas9-LSD1, dCas9-VP64, and dCas9-TET1CD fusion constructs in vitro[28–30,43,44]. Unlike our approach those studies did not use an all-in-one fusion protein but two individual components, which are only functional when both are delivered and expressed simultaneously in the same cell, thus thereby limiting their utility in vivo.

Our study is further in line with a very recent report, where a demethylating system based on dCas9 is fused to the repeating peptide GCN4, which recruits an anti-GCN4 single-chain variable fragment (scFv) fused to the effector domain of TET1CD[45]. This system was successfully introduced into the embryonic mouse brain by in utero electroporation and thereby reactivated expression of specific genes including Gfap in vivo. Our application is in contrast by lentiviral delivery and is made possible through a considerably smaller size of the construct in our study

as compared to constructs utilized in that study and thus presents a more feasible therapeutic approach in vivo as compared to electroporation-based gene delivery. Thus, although there are previous reports with respect to gene-specific demethylation both in vitro and in vivo, to our knowledge, this study is the first to describe an effective CRISPR-based epigenetic therapy in a disease model. Unlike all previously published reports where only catalytic domain of TET1 or TET2 have been fused with DNA-binding domains ZNF, TALE, or dCas9[25,26,45], to our knowledge

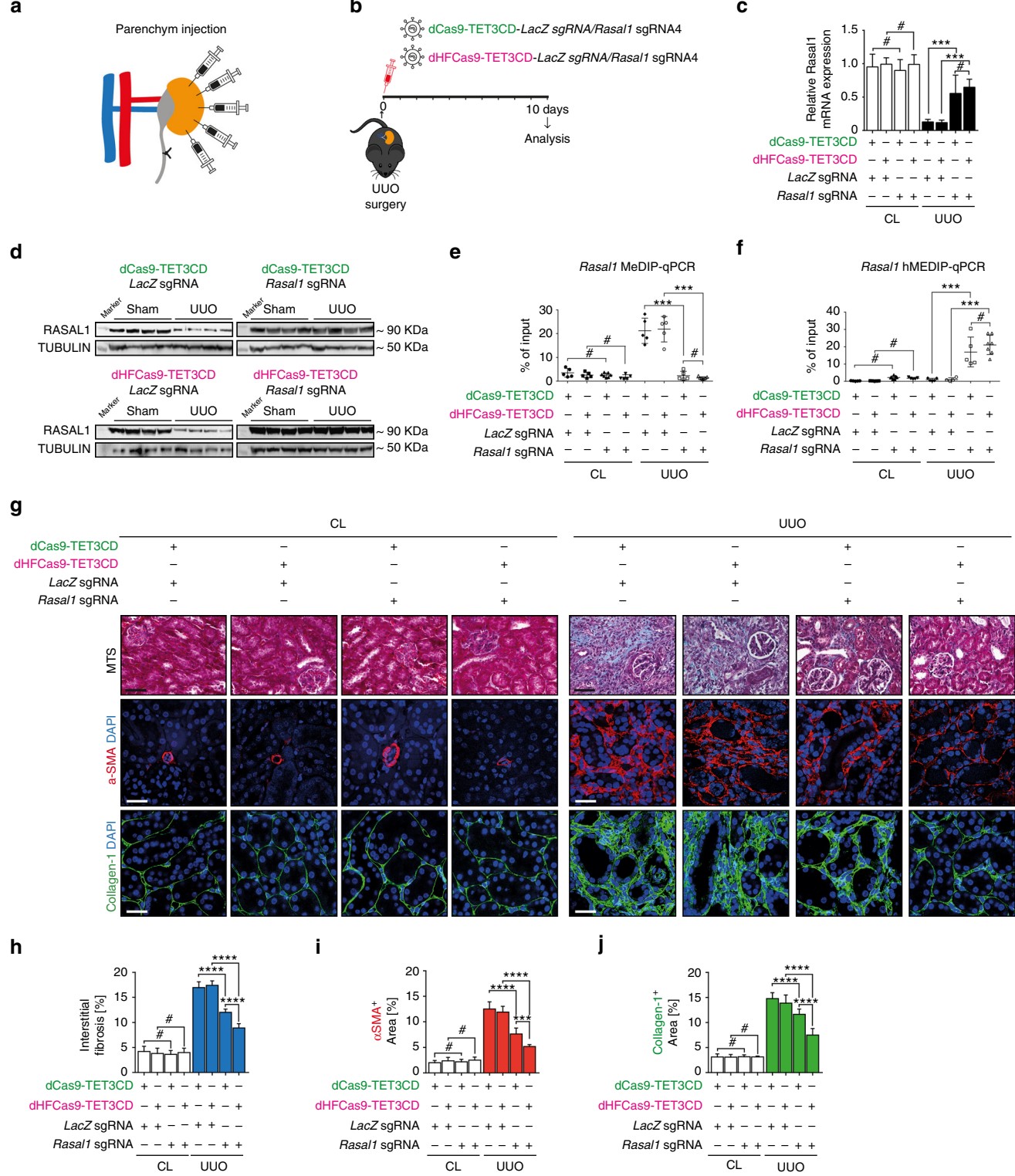

we are the first to validate gene-specific demethylation by TET3CD by the use of dCas9-TET3CD.

Even more importantly, none of the previous studies used a high-fidelity Cas9. Off-target effects largely hinder the utility of CRISPR/Cas9 technology therapeutically. Through substitution of four amino acids, Kleinstiver and his colleagues developed a high-fidelity Cas9, which abolishes the redundant energetics of SpCas9-sgRNA with the consequence of diminished off-target effects[41]. Thus, as long as number of off-target genes is kept low through the use of HFCas9, adverse effects of our strategy in cells other than the target cells are unlikely since only methylated genes which were originally actively transcribed within target cells can be reactivated. This provides a conceptual advantage over other therapies based on e.g. AAV-delivered overexpression constructs where the targeting of specific cell types still remains a challenge.

## Methods

**Plasmids**. The sgRNA sequences (*RASAL1*, *EYA1*, *LRFN2*, *KL*, *Rasal1*, and *Kl*) were designed by the online tool Blueheronbio (Origene, Herford, Germany). The control *LacZ* sgRNA sequence was the same as previously used[38]. The sgRNA sequences (Supplementary Table 1) were inserted into the pLenti-Cas-Guide plasmid (Origene, Herford, Germany) with BamHI and BsmBI restriction sites to generate pLenti-Cas-sgRNA (*RASAL1*, *EYA1*, *LRFN2*, *KL*, *Rasal1, and Kl*) constructs and confirmed by DNA sequencing. The wild-type Cas9 open reading frame was removed from the vector with Age1 and Not1 restriction sites. The plasmids encoding H840A SpCas9 and encoding high-fidelity SpCas9-HF4 were gifts from Jennifer Doudna and Keith Joung, respectively. (Addgene plasmids #39316[46] and #72249[41].) The pLenti-dCas9-sgRNA gene demethylation constructs were generated by cloning H840 SpCas9 in frame into the digested pLenti-sgRNA vectors by PCR using Phusion high-fidelity DNA polymerase (NEB, Ipswich, USA) with Age1 (5′) and Xba1, Not1 (3′) restriction sites with two NLS (nuclear localization signal) peptides at the N- and C-terminus each with a primer pair that introduced the D10A mutation[30] (Supplementary Table 6). The TET3CD (catalytic domain, aa 850- 1795)[24] was amplified from a human TET3 ORF (Origene, Herford, Germany) with a primer pair which introduced Age1, a start codon, Xba1, a Gly- Gly-Gly- Ser- Gly linker (5′), and Not1 (3′) and then inserted into the digested pLenti-dCas-LacZ vector to generate pLenti-dCas9-TET3CD-LacZ. The sequence and the coding frame for dCas9 and for TET3CD were confirmed by DNA sequencing and by western blot (Supplementary Fig. 1). The final constructs pLenti-dCas9-TET3CD-sgRNA (*RASAL1*, *EYA1*, *LRFN2*, *KL*, *Rasal1*, and *Kl*) were generated by removing TET3CD from pLenti-dCas9-TET3CD-LacZ, subsequently inserted into pLenti-dCas9-sgRNA (*RASAL1*, *EYA1*, *LRFN2*, *KL*, *Rasal1*, and *Kl*) with Xba1 and Not1 restriction sites. The primer sequences used for PCR cloning are listed in Supplementary Table 2.

**DNA isolation, MeDIP, and hMeDIP assay**. Animal tissues or cell pellets were lysed by DNA lysis buffer (Qiagen, Hilden, Germany) and precipitated and purified using DNeasy Blood & Tissue Kit (Qiagen, Hilden, Germany) according to the manufacturer's protocol. Prior to immunoprecipitation, genomic DNA was

sonicated (Qsonica, Newtown, USA) to produce DNA fragments ranging in size from 200 to 1000 bp, with a mean fragment size of around 300 bp. Methylated DNA was captured using Methylamp Methylated DNA Capture Kit (Epigentek, Farmingdale, USA). In total 1.0 μg of fragmented DNA was applied in every antibody-coated well and incubated at room temperature on a horizontal shaker for 2 h. The immunoprecipitated DNA was released by proteinase K. The DNA was eluted and adjusted to a final volume of 100 μl with nuclease-free water. For each sample, an input vial was included using total sonicated DNA as loading control. Hydroxymethylated DNA was captured using EpiQuick Hydroxymethylated Immunoprecipitation (hMeDIP) Kit (Epigentek, Farmingdale, USA) according to the manufacturer's protocol. A volume of 0.5 μg of sonicated DNA was added to each antibody-coated well and incubated at room temperature on a horizontal shaker for 90 min. The immunoprecipitated DNA was released by proteinase K. The DNA was eluted and diluted to a final volume of 200 μl with nuclease-free water. For each sample, an input vial was included using total sonicated DNA as loading control. The primer sequences used for MeDIP/hMeDIP-qPCR cloning are listed in Supplementary Table 3.

**Glucosylation-mediated restriction enzyme sensitive PCR**. The EpiMark Kit (NEB, Ipswich, USA) was used to validate the conversion from 5′mC to 5′hmC at the selected *RASAL1* promoter region. The assay was performed according to the manufacturer's protocol. Briefly, 10 μg of genomic DNA was used and equally divided into two reactions, one treated with T4-phage βGT at 37 °C for 12 h, the other one was kept as untreated control. Both the βGT-treated and untreated samples were then divided into three PCR tubes and digested with either MspI, HpaII, or left uncut at 37 °C for an additional 12 h. Samples were treated with proteinase K at 40 °C for 10 min prior to dilution to a final volume of 100 μl with nuclease-free water and heating to 95 °C for 5 min. PCR was carried out in a volume of 5 μl for each sample on a PCR cycler (Eppendorf, Hamburg, Germany) with a standard PCR program. The primer sequences used for PCR are listed in Supplementary Table 4. To visualize the PCR products, samples were loaded into the Bioanalyzer 2100 electrophoresis system (Agilent Technologies, California, USA). Electrophoresis results are shown as a virtual gel as previously described[47].

**Bisulfite sequencing**. Purified cellular DNA was bisulfite-treated using the EZ DNA Methylation-Lightning Kit (Zymoresearch, Irvine, USA) according to the manufacture's protocol. To amplify the *Rasal1* and *Kl* promoter fragments, a touchdown PCR program was performed using Taq DNA Polymerase (Sigma-Aldrich, St. Louis, USA). The first round of PCR consisted of the following cycling conditions: 94 °C for 2 min, 6 cycles consisting of 30 s at 94 °C, 30 s at 60–55 °C (reduce 1 °C after each cycle), and 30 s at 72 °C. The second round of PCR consisted of the following cycling conditions: 32 cycles consisting of 30 s at 94 °C, 30 s at 55 °C, and 30 s at 72 °C. The final elongation consisted of 72 °C for 6 min. The sequences of the PCR primers are listed in Supplementary Table 5. The PCR products were purified using the QIAEX II Gel Extraction Kit (Qiagen, Hilden, Germany), cloned into the pGEM-T Vector (Promega, Wisconsin, United States) and transformed into Top10 Competent E.coli Cells (Life Technologies, Carlsbad, USA). The plasmid DNA was then purified with DNA Plasmid Miniprep Kit (Qiagen, Hilden, Germany) and sequenced (Seqlab, Göttingen, Germany).

**Cell culture and transfection**. TK173 and TK188 kidney fibroblasts were isolated from human kidney biopsies[4]. HEK293 and HK2 cells were purchased from ATCC (Teddington, UK). Mouse kidney fibroblasts were generated in our lab. Murine

**Fig. 5** Targeted *Rasal1* promoter demethylation by dCas9/dHFCas9-TET3CD-*Rasal1*-sgRNA fusion protein ameliorates kidney fibrosis. **a** Schematic showing the parenchymal injection of lentiviral particles containing dCas9/dHFCas9-TET3CD fusion protein into UUO-challenged kidneys. **b** Schedule of UUO mouse surgery, lentivirus injection, and analysis. **c** qRT-PCR results showing that Rasal1 mRNA expression was significantly induced in UUO-challenged kidneys transduced with dCas9/dHFCas9-TET3CD-*Rasal1*-sgRNA, but not in UUO-challenged kidneys transduced with control dCas9/dHFCas9-TET3CD-*LacZ*-sgRNA. There is no significant difference between dCas9-TET3CD and dHFCas9-TET3CD constructs. Results were normalized to reference gene Gapdh (expression is presented as mean value; error bars represent S.D., $n \geq 5$ in each group, # not significant, ***$p < 0.001$). **d** Western blots showing restored RASAL1 protein expression in UUO-challenged kidneys which were transduced with lentivirus expressing dCas9/dHFCas9-TET3CD-*Rasal1*-sgRNA. The membranes were restriped and re-probed with α-TUBULIN antibody to serve as equal loading control. **e, f** UUO-challenged kidneys which were transduced with lentivirus expressing dCas9/dHFCas9-TET3CD-*Rasal1*-sgRNA show significantly reduced *Rasal1* promoter methylation by MeDIP-qPCR assay (**e**) and increased hydroxymethylation by hMeDIP-qPCR assay (**f**). There is no significant difference between dCas9-TET3CD and dHFCas9-TET3CD constructs. The results were calculated relative to input. The data are presented as mean value, error bars represent S.D., $n \geq 5$; # not significant, ***$p < 0.001$. **g** Kidney sections from UUO- and sham-operated mice which were transduced with lentivirus expressing dCas9/dHFCas9-TET3CD-*Rasal1*-sgRNA or dCas9/dHFCas9-TET3CD-*LacZ*-sgRNA were stained for Masson's trichrome (MTS) (representative light microscopy images are shown in the top row), Collagen-1 or α-SMA (representative confocal images are shown in the middle and bottom row, respectively) (Scale bars: 25 μm or 50 μm). **h–j** Quantification of the percentage of total interstitial fibrosis and immunostained positive cells in each group is depicted (data are presented as mean value, error bars represent S.E.M., $n \geq 5$ in each group, # not significant, ***$p < 0.001$, ****$p < 0.0001$). Both dCas9-TET3CD and dHFCas9-TET3CD lentivirus transduced UUO-operated kidneys show significantly decreased interstitial fibrosis level and a significantly decreased number of α-SMA- and Collagen-1-positive cells. HFCas9-TET3CD shows significantly better efficacies when compared to dCas9-TET3CD

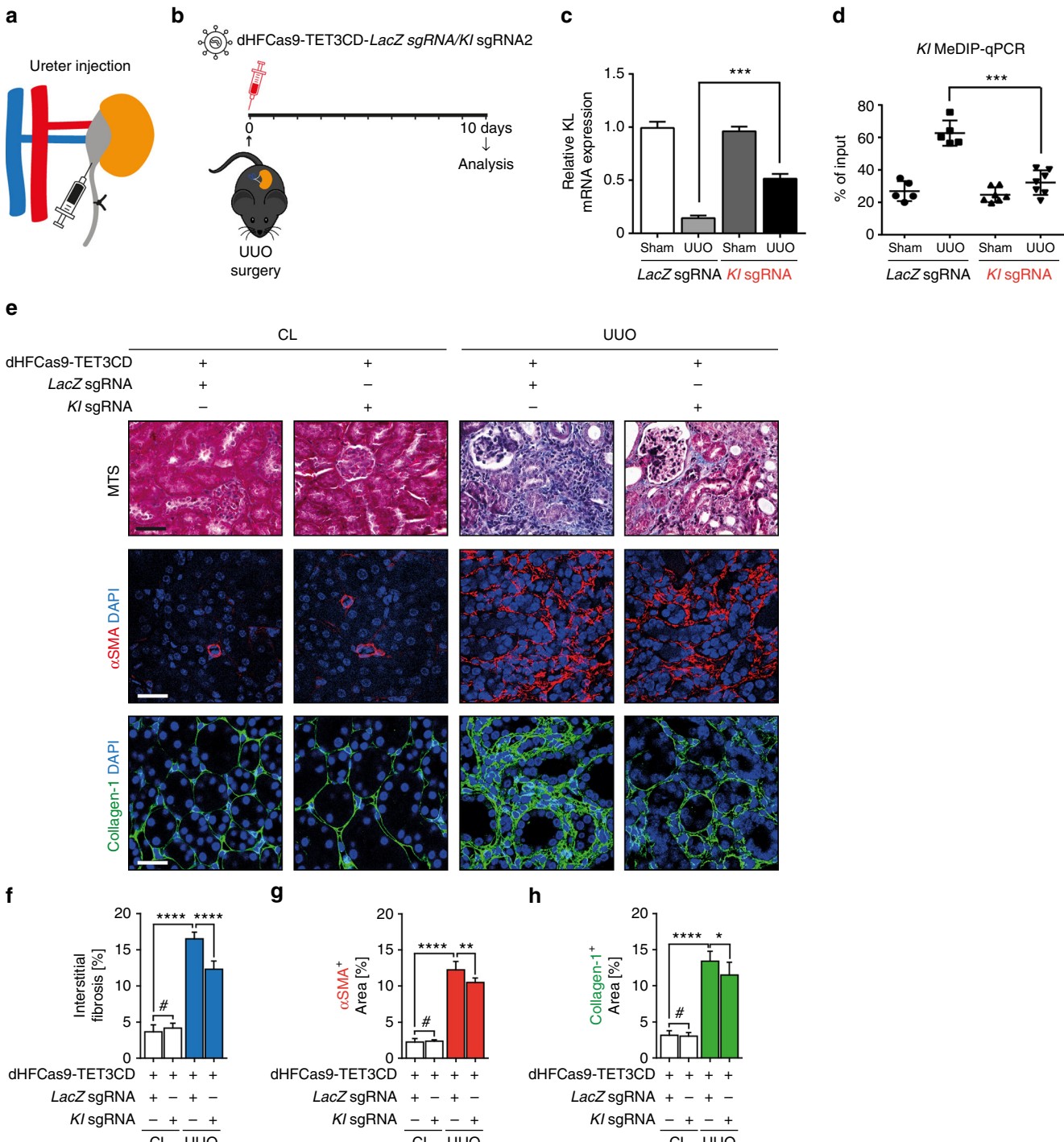

**Fig. 6** Induction of *Klotho* promoter hydroxymethylation by dHFCas9-TET3CD-*Klotho*-sgRNA in tubular epithelial cells ameliorates kidney fibrosis.
**a** Schematic shows retrograde ureter injection of lentiviral particles containing dHFCas9-TET3CD-*Klotho*-sgRNA into UUO-operated kidneys. **b** Schedule of UUO mouse surgery, lentivirus injection, and analysis. **c** qRT-PCR results showing that Kl mRNA expression was significantly induced in UUO-operated kidneys transduced with dHFCas9-TET3CD-*Kl*-sgRNA but not in UUO-challenged kidneys transduced with *LacZ*-sgRNA control. Results were normalized to reference gene Gapdh (expression is presented as mean value, error bars represent S.E.M., *n* = 6 in each group, ***p* < 0.001). **d** UUO-operated kidneys which were transduced with lentivirus expressing dHFCas9-TET3CD-*Kl*-sgRNA show significantly reduced *Kl* promoter methylation level by MeDIP-qPCR assay. The results were calculated relative to input. The data are presented as mean value, error bars represent S.D., *n* ≥ 5, ***p* < 0.001. **e** Kidney sections from UUO- and sham-operated mice which were transduced with lentivirus expressing dHFCas9-TET3CD-*Kl*-sgRNA or *LacZ*-sgRNA were stained for Masson's trichrome (MTS) (representative light microscopy images are shown in the top row), Collagen-1 or α-SMA (representative confocal images are shown in the middle and bottom row, respectively) (Scale bars: 25 μm or 50 μm). **f–h** Quantification of the percentage of total interstitial fibrosis and immunostained positive cells in each group are depicted (data are presented as mean value, error bars represent S.E.M., *n* ≥ 5 in each group, # not significant, **p* < 0.05, ***p* < 0.01, *****p* < 0.0001). UUO-challenged kidneys transduced with lentivirus expressing dHFCas9-TET3CD-*Kl*-sgRNA show significantly decreased interstitial fibrosis level and a significantly decreased number of α-SMA- and Collagen-1-positive cells

MCT cells were kindly provided by Dr. Eric G. Neilson (Northwestern University, Evanston, IL). Human TK188, HK2, HEK293, 293 T, MCT, and mKF (passages between 3 and 5) were cultured in DMEM (Gibco, Carlsbad, USA) supplemented with 2 mmol/l L-glutamine, 100 g/ml penicillin, 100 g/ml streptomycin and 10% heat-inactivated fetal bovine serum (FBS, Cellgro, Manassas, USA) at 37 °C in 5% $CO_2$. For transfection experiments, cells were pre-plated and cultured overnight and transfected with Lipofectamine 2000 (Life Technologies, Carlsbad, USA) according to the manufacturer's protocol. Briefly, the plasmid DNA (2.5 µg each) and Lipofectamine 2000 were mixed in a ratio of 1:2 in a total volume of 500 µl of Opti-MEM (Life Technologies, Carlsbad, USA) and incubated at room temperature to form complexes for 20 min. The transfection complex was added to the cells in basic medium without serum. After overnight incubation, the medium was replaced back to complete growth medium.

**Chromatin immunoprecipitation-next genomic sequencing.** ChIP assay was performed using 1 Day ChIP kit and Shearing ChIP kit (Diagenode, Denville, USA) according to the manufacturer's protocol. After lentiviral transduction and cross-linking, mouse kidney fibroblasts were fixed with formaldehyde and the DNA was sheared into small fragments. After incubation with a Myc-tag antibody (Cell signaling, Beverly, USA), the pulled-down complexes were de-crosslinked and treated with proteinase K. The purified DNA samples were proceeded further for library preparation by TruSeq RNA Library Prep Kit v2 (Illumina, USA) and quality control and library validation were performed by Fragment Analyzer and Kapa PCR (Illumina, USA), respectively. The fragments were sequenced by an Illumina HiSeq4000 instrument. For data analysis, a previous established protocol[48] was followed. Briefly, the raw reads from two independent biological replicates were first concatenated and then peak calling was performed with MACSII in order to obtain the count numbers. We used settings for narrow peaks (200 bp window size, 200 bp gap size, and false discovery rate of 0.01) in all cases.

**Virus packaging and titration.** Lentiviruses were produced using 293 T virus packaging cells upon transfection with a combination of 2nd Generation Packaging System Mix (Abmgood, Richmond, Canada), pLenti-dCas9-TET3CD-sgRNA (RASAL1, EYA1, LRFN2, KL, Rasal1, and Kl), and Lentifectin (Abmgood, Richmond, Canada) according to the manufacture's protocol. Lentiviral supernatant was collected 2 and 3 days after transfection, filtered through a 0.45 µm filter and concentrated with Lenti-X Concentrator (Clonetech, Heidelberg, Germany). The viral particles were resuspended in PBS and stored at −80 °C. Lentivirus titration was determined by the Lenti-X qRT-PCR Titration Kit (Clonetech, Heidelberg, Germany) and 8 µg/ml Polybrene (Sigma-Aldrich, Munich, Germany) was added to the viral solution for the in vivo and in vitro transduction experiments.

**Rasal1 mutant mouse production.** Two different mouse embryonic stem cell clones (A03, H03) carrying targeted gene knockout alleles were received from EUCOMM international knockout mouse consortium. Stem cells were from the JM8 line on a C57BL/6 N genetic background reference. The targeted $Rasal1^{tm1a}$ (KOMP)WTSI allele carried a gene-trap DNA cassette, inserted into the second intron of the gene, consisting of a splice acceptor site, an internal ribosome entry site, and a β-galactosidase reporter. The use of the splice acceptor site is purposely to generate a truncated non-functional transcript. After confirming the correct genome targeting by long range PCR (Supplementary Fig. 12a), Rasal1 mutant ESCs were injected into blastocysts to generate chimera mice which were further bred with C57BL/6 N mice to generate $Rasal1^{tm1a/+}$ mice. Homozygous $Rasal1^{tm1a/tm1a}$ mice were then generated by inbreeding (Supplementary Fig. 12b). Copy number variation analysis of Rasal1tm1a mouse kidneys confirmed that mutant mice carried the correct copy number of gene-trap DNA cassette (Supplementary Fig. 12c). The sequence of primers used for characterizing the $Rasal1^{tm1a/+}$ mice are listed in Supplementary Table 6. Mice were euthanized at 12–14 weeks of age under isoflurane anesthesia with cervical dislocation. Tissues were rapidly harvested and quick-frozen in liquid nitrogen, and stored at −80 °C. All animal work followed the Guide of the Institutional Review Board of the University of Göttingen and the responsible government authority of Lower Saxony (Germany).

**Unilateral ureteral obstruction.** All animal experiments complied with ethical regulations and were conducted according to the animal experimental protocols which were approved by the Institutional Review Board of the University of Göttingen and the responsible government authority of Lower Saxony (Germany). Eight-to-twelve-week-old wild-type C57BL/6 N mice were used for the study. After anesthesia with isoflurane inhalation, analgesia was performed by subcutaneous Buprenorphine injection. The ureter was separated from the surrounding tissues and the left ureter was clamped distal to the infusion site by two ligatures[14]. The abdominal muscles were sutured with absorbable suture, and the skin was closed with non-absorbable suture[49,50]. Mice were sacrificed 10 days after ureter ligation and viral solution injection. The UUO-operated and the contralateral kidney were removed for histological analysis.

**Intrarenal artery/vein infusion.** After anesthesia, the aorta, inferior vena cava, and the right renal vessels were visualized through a midline incision. After completing the UUO surgery, the left renal artery/vein was clamped and the renal

artery/vein was cannulated with a 27-gauge needle. Either control LacZ-sgRNA or Rasal1/Kl-sgRNA lentiviral solution (80 µl about $1 \times 10^8$ TUs) was slowly injected under gentle pressure to avoid leaking of the injection solution. As the needle was removed, a clamp was placed around the renal artery to trap the solution in the kidney. Blood flow was stopped for 15 min and then the clamps were removed, thus restoring blood flow to the kidney.

**Retrograde ureteral infusion.** Upon completion of the UUO surgery, the left ureter-pelvic junction was cannulated with a 27-gauge needle. Viral solution (80 µl about $1 \times 10^8$ TUs) was infused under gentle pressure to avoid renal pelvic distention and leaking of the viral solution.

**Intraparenchymal injection.** Once completing the UUO surgery, the left kidney was locally injected with viral solution (80 µl about $1 \times 10^8$ TUs in total) into four different sites.

**Histology and Immunofluorescence.** Paraffin-embedded kidneys were sectioned at 3 µm and Masson's Trichrome Stain (MTS) was performed. We assessed the fibrotic area using CellSens (Olympus, Tokyo, Japan) software, as previously described[14]. For immunofluorescent staining, primary antibodies against α-smooth muscle actin (1:100 diluted, Abcam, Cambridge, UK), Collagen-1 (1:100 diluted, Abcam, Cambridge, UK), and Collagen-4 (1:100 diluted, BD/Pharmingen, San Diego, USA), and Alexa Fluor 488, 568 (1:300 diluted, Life Technologies, USA, Carlsbad, USA) secondary antibodies were used. Nuclear staining was performed using 4′, 6-diamidino-2-phenylindole (DAPI, Vector Labs, Burlingame, USA). Relative areas positive for α-SMA and Collagen-1 per visual fields were analyzed at magnification × 40.

**RNA extraction, cDNA synthesis, and real-time PCR analysis.** Animal tissues were shredded by TissueLyser LT (Qiagen, Hilden, Germany). Total RNA was extracted from the shredded tissues or cells by direct lysis with TRIzol reagent (Life Technologies, Carlsbad, USA) and RNA isolation was performed using the Pure-Link RNA Mini Kit (Life Technologies, Carlsbad, USA) according to the manufacturer's protocol. For first-strand cDNA synthesis, 1 µg of total RNA was treated with DNase I (Sigma-Aldrich, Munich, Germany) and then converted into complementary DNA (cDNA) using the SuperScript II System (Life Technologies, Carlsbad, USA). For qRT–PCR analysis, 2 µl of diluted cDNA (1:10) as a template and the Fast SYBR Green Master Mix (Life Technologies, Carlsbad, USA) were used in a final volume of 20 µl for each reaction. Real-time PCR was performed in triplicate in a 96-well format by StepOne Plus Real-Time PCR system (Life Technologies, Carlsbad, USA). The real-time PCR primers are listed in Supplementary Table 7. The relative expression levels were standardized to GAPDH using $2^{-\Delta\Delta Ct}$ methods.

**Western blot.** Animal tissues were shredded by TissueLyser LT (Qiagen, Hilden, Germany). The shredded tissues and cells were lysed in NP40 buffer (Life Technologies, Carlsbad, USA) containing protease inhibitor cocktail (Roche, Mannheim, Germany). Lysates were mixed with loading buffer and heated at 95 °C for 5 min. Protein samples were resolved on 4–12% SDS–PAGE gels (Life Technologies, Carlsbad, USA) and transferred onto nitrocellulose membranes (GE Healthcare, Freiburg, Germany). Non-specific antibody binding was blocked with 5% nonfat milk in TBST buffer (50 mM Tris/150 mM NaCl/0.1% Tween-20) for 1 h. The membranes were incubated with primary antibodies in incubation solution (2% milk in TBST) overnight. Myc-tag antibody (Cell Signaling #5605, Danvers, USA) was diluted 1:2500; Tubulin antibody (Sigma-Aldrich #T5168, Munich, Germany) was diluted 1:5000; RASAL antibody (Abcam #ab168610, Cambridge, UK; Biorbyt #orb101674, Cambridge, UK) was diluted 1:1000. The membranes were washed three times with 2% milk in TBST and incubated with HRP conjugated secondary antibody for 1 h. Membranes were visualized using LumiGLO chemiluminescence (Cell Signaling, Danvers, USA) and images were documented by a ChemiDoc MP System and processed using ImageLab software (Bio-Rad, Munich, Germany). All uncropped blots are included in Supplementary Fig. 13.

**Statistical analysis.** All qRT-PCR data for RNA expression analysis are presented as the mean ± SD and represent a minimum of three independent experiments. As for MeDIP-qPCR, two biological replicates with three technical replicates each were analyzed, and the average of six values was used for further statistical analysis. Histological analysis was assessed as 10 visual fields per mouse and average values of each mouse were calculated. The average values for each mouse ($n = 6$) was then used to calculate average per group. Statistical analysis was then performed on these averages using One-way ANOVA with Bonferroni post-hoc analysis comparing selected pairs of columns.

**Data availability.** The authors declare that all relevant data supporting the findings of this study are available within the paper and its supplementary information files. All the figures and graphics were created by the authors. Complete ChIP-sequencing data has been deposited in BioProject https://submit.ncbi.nlm.nih.gov/subs/bioproject/SUB3776305/biosample with accession number [SUB3776305].

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

## Acknowledgements

This work was supported by DFG grants SFB1002/C01 (to E.M.Z.) and DFG ZE523/4–1, SFB1002/D03 (to M.Z.) and funds from the University of Göttingen Medical Center (UMG) to E.M.Z. and M.Z. X.X. received support from the "seed funding research program" of the Faculty of Medicine, Georg August University Göttingen and postdoc start-up grant, DZHK. This work was further supported by the German Ministry of Research and Education (BMBF) through the DZHK (German Centre for Cardiovascular Research). The authors thank Dr. Wanhua Xie for helping with ChIP-sequencing data analysis and Annika Faust, Sarah Rinkleff, and Anika Krueger for technical assistance.

## Author contribution

X.X. conceptual design, experimental design, data collection, data analysis, and wrote draft. X.T. experimental design, data collection, data analysis, and edited manuscript. B.T. designed and performed animal experiments, data collection, and analyzed data. S.S., T.W. data collection. M.H. data collection, edited manuscript. T.M. provided necessary assistance with S2 animal studies. R.K. and G.H. proof reading and conceptual advice. M.Z. and E.Z. conceptual and experimental design and wrote and edited the manuscript.

## Additional information

**Competing interests:** X.X. and E.Z. are inventors on a patent entitled "Method for re-expression of different hypermethylated genes involved in fibrosis, like hypermethylated RASAL1 and use thereof in treatment of fibrosis as well as kit of parts for re-expression of hypermethylated genes including RASAL1 in a subject" (patent number EP2018/054619). The remaining authors declare no competing interests.

