## [Peer Review File · Nature Communications]

Reviewers' comments:

Reviewer #1 (Remarks to the Author):

Comments to the authors

NCOMMS-16-22492-T

Gene-specific hydroxymethylation via CRISPR/Cas9-TET3 rescues expression of methylated RASAL1 and attenuates experimental fibrosis.

TET enzymes catalyse hydroxymethylation of methylated promoter CpGs, leading to DNA demethylation and gene re-expression. Xu et al. generated a dCas9 (cleavage-defective Cas9)-TET3 targeted to specific genes by a guiding RNA.

They target the promoter of RASAL1 by using dCas9-TET3CD guided by RASAL1 sg-RNA, resulting in rescue of RASAL1 expression in fibrotic human kidney fibroblasts in vitro. They also report a lentiviral-based delivery approach to express this system in a mouse model of kidney fibrosis, which results in Rasal1 expression.

There are several published works which use dCas9 and TET to demethylate a specific gene. However, application of the technology to a therapy model could be significant if therapeutic effect is strong.

In the current study, therapeutic effect is weak and several points seem weak to me. Therefore, I think the manuscript is not strong enough in its current format to be published in Nature Communications.

1. Although there are many papers describing dCas9-TET1, there is no previous paper describing dCas9-TET3. Therefore, it is important to validate this method. One important control is missing to rule out the possibility that this effect is due to overexpression of TET3. This possibility can be excluded by the experiments using catalytically-defective TET3 and sgRNA4.
2. Because this study aimed the therapeutic application, it is very important to validate the safety of the method. CRISPR/Cas9 system is thought to have off-target effects. Therefore, it is important to perform off-target analysis. In addition, TET3 could change the genome-wide methylation. In this study, off-target analysis, genome-wide methylation analysis, and genome-wide expression analysis were not performed. Only 7 genes were analyzed by RT-PCR. It is essential to perform genome-wide analysis (BS-seq, RNA-seq) and off-target analysis.
3. Although expression of Rasal1 is significantly induced in mouse model, improvement of fibrosis is very small. It suggests that genes other than Rasal1 could be involved in generation of the

disease. In this point, it is important to examine whether Rasal1 knockout mouse show the fibrosis phenotype.

4. Although five sgRNAs were used for Rasal1, only sgRNA4 can demethylate the gene, suggesting that this method can be applied to only limited locus. To test whether this method can be generally applied to genes, it is important to perform systematically application of dCas9-TET3 in vivo is important.

5. The authors argue that TET3CD can demethylate longer range compared to TET1CD because TET3CD have larger domain than TET1CD. However, they did not show the range of demethylation using TET1CD in this locus. Because the range of demethylation could be different between loci, they should perform the experiments for several loci before generalize the discussion.

6. The authors argue that TET3CD can demethylate longer range compared to TET1CD because TET3CD have larger domain than TET1CD. This line of investigation would be helpful to the scientific community. Unfortunately, there is no biochemical evidence for this. It would be better to show some biochemical evidence for this.

Reviewer #2 (Remarks to the Author):

This is technically well performed methods study that will be of interest to investigators wanting an approach for modifying DNA methylation in a sequence specific way.

However, it is not an entirely novel study since there are already recent publications reporting the use of CRISPR/Cas9 technology to modify methylation in a site specific manner, including demonstration that the technology works in vivo. Examples include Volta et al, NAR March 2016; Xu et al, Cell Discovery 2, 2016; and Shawn Liu et al in Cell Sept 2016. Hence there is a question as to whether the reporting of this technology is now sufficiently novel for the readership of Nature Communications.

Also it is already established that RASAL1 is important in renal fibrosis and that its expression is controlled by DNA methylation, this information is nicely confirmed in the present study but again lacks novelty. The effects of targeted hydroxymethylation of RASAL1 were however modest and were only examined in one experimental model of fibrosis, this therefore limits the interest of the work to the wider fibrosis research community.

The mechanisms by which hydroxymethylation altered RASAL1 expression were not fully examined and as such this work does not sufficiently advance our understanding of the molecular regulation of RASAL1 or in turn the mechanisms by which RASAL1 regulates fibrosis.

Reviewer #3 (Remarks to the Author):

This is a very interesting manuscript about the design of a novel fusion protein using dead Cas9 fused to the catalytic domain of the TET3 enzyme to enable site specific promoter demethylation. The authors provide many controls to show specificity of their approach, replicate the approach in human and mouse, and then demonstrate proof of principle in the UUO model of kidney injury and fibrosis.

Overall, I think this is a fabulous Methods paper, demonstrating proof of principle and should be of appeal to the readership. The effect size on mouse kidney disease biology is very small, and of uncertain significance. Nevertheless, as a Methods paper, it will be of value to the community.

In addition, I have a number of significant concerns that should be addressable, to enhance the appeal and digestibility of the complicated biology.

1. It appears that the manuscript was prepared for Nature, with 3 figs and 18 supp figs. It is very hard to read in places, with poor explanation of terms, and concepts for a general reader, for whom Nat Comm is directed. For example bisulfite sequencing is poorly explained, hMeDIP is poorly explained, TK188 cells are poorly described, Fig 1B is poorly explained and defined. IN other places terms and explanations for approach are poorly explained to a non-specialist reader. I would propose that the text length is extended and some of the supp figures could be brought into the main text.

2. A diagram explaining the promoter, showing methylation and hydroxymethylation would be very helpful to the reader.

3. It would be helpful to demonstrate a lack of off target effect for the approach. So using specific guide4 and the fusion protein, can you show a lack of off target effects, at candidate off target sites in the genome?

4. The text repeatedly states that RASAL1 methylation is causal in fibrotic disease, but since it is a DNA state triggered by disease and cytokine milieu, it could not be considered causal. It is in response to disease, not causal of disease. The use of this term should be modified, as it is misleading.

4. There is a large discrepancy between the effect of the approach shown in Fig 1E, 1L, and in 2G and 2H when compared to the apparent restoration of protein levels showing in Fig 2F. It would be more convincing to increase the number of replicates in Fig 2F so that the degree of protein restoration can be more clearly defined.

5. Is there an explanation for why there is no endothelial or epithelial uptake of the viral reporter for sgRNA in Fig 3B. It seems unlikely that the virus does not get into other cell types and generate the fusion protein. Note DDK is not defined

6. The effect size of this approach on hard outcome measures, fibrosis, collagen I and α SMA is very small. I doubt that the findings are statistically significant at the $P < 0.0001$ level as claimed in Fig 3K-M. Given the limited utility of the UUO model, would it be worth considering a more chronic model to determine whether a larger effect can be achieved, such as the folate model the authors have used previously?

Reviewer #1

TET enzymes catalyse hydroxymethylation of methylated promoter CpGs, leading to DNA demethylation and gene re-expression. Xu et al. generated a dCas9 (cleavage-defective Cas9)-TET3 targeted to specific genes by a guiding RNA.

They target the promoter of RASAL1 by using dCas9-TET3CD guided by RASAL1 sg-RNA, resulting in rescue of RASAL1 expression in fibrotic human kidney fibroblasts in vitro. They also report a lentiviral-based delivery approach to express this system in a mouse model of kidney fibrosis, which results in Rasal1 expression.

There are several published works which use dCas9 and TET to demethylate a specific gene. However, application of the technology to a therapy model could be significant if therapeutic effect is strong.

Thank you for your positive assessment of our work. Our study is indeed the first to apply gene-specific demethylation therapeutically.

In the current study, therapeutic effect is weak and several points seem weak to me. Therefore, I think the manuscript is not strong enough in its current format to be published in Nature Communications.

In response to the reviewer's comment we have now performed extensive experiments to systematically advance our technology in the revised version of the manuscript, with improved therapeutic effect as outlined below.

1. Off-target effects are imminent to the dCas9 system. Therefore, in order to assess what factors could impede therapeutic effect, we now performed a genome-wide search for off-target effects of the dCas9-TET3CD system used in our previous study, which revealed that several pro-fibrotic genes were also targeted (including Anxa4 and Nlrp5, see **Table 1**), which likely counteracted the therapeutic effect. We therefore now developed an endonuclease deactivated high-fidelity Cas9 (dHFCas9) fused to TET3 catalytic domain instead of dCas9. This reduced off target genes by 85% (**Figure 2g-j**) and now does not target Anxa4 or Nlrp5 (**Table 1**).

2. We have now systematically compared different lentiviral delivery methods to optimize the therapeutic effect. Through the use of RFP and GFP labelled lentiviruses we found that among 4 injection sites (into renal vein, into renal artery, into renal parenchyma, as well as retrograde into the ureter), local parenchymal injection was most effective to target interstitial cells (**Figure 4**).

Using this modified and improved technology, kidney fibrosis was now ameliorated by **48,7%** through intraparenchymal injection of dHFCas9-TET3CD-Rasal1-sgRNA as compared to LacZ control vector (**Figure 5g-j**).

1. Although there are many papers describing dCas9-TET1, there is no previous paper describing dCas9-TET3. Therefore, it is important to validate this method. One important control is missing to rule out the possibility that this effect is due to

overexpression of TET3. This possibility can be excluded by the experiments using catalytically-defective TET3 and sgRNA4.

This is a very good point. As per the reviewer's suggestion, we have now generated a catalytically inactive TET3 catalytic domain fused with dCas9 and different sgRNAs (dCas9-TET3CDi-sgRNA) and compared it to active TET3 catalytic domain fused with dCas9 and sgRNAs (dCas9-TET3CD-sgRNA). We compared inactive to active TET3 catalytic domain for three different human genes (*RASAL1*, *EYA1* and *LRFN2*), which are hypermethylated in fibrotic fibroblasts (**Figure 1a-h**) and for *KLOTHO*, which is hypermethylated in human tubular epithelial cells upon TGF β 1 treatment (**Figure 1i-k**). For all four genes the constructs containing catalytically inactive TET3 catalytic domain showed no effect on gene expression (**Figure 1 c-e, i**) and (unlike active TET3 catalytic domain), did not impact methylation levels (**Figure 1f-h, j-k**). As an additional control, we also performed overexpression experiments of TET3 catalytic domain (TET3CD) alone in fibrotic fibroblasts TK188 and in tubular epithelial cells HK2 upon TGF β 1 treatment, and did not detect an effect on gene expression in any of the four genes (**Supplementary Figure 3**). These important controls further corroborate that the observed effect of dCas9/dHFCas9-TET3CD-sgRNAs is due to gene-specific demethylation mediated by TET3 through gene-specific guiding RNAs.

2. Because this study aimed the therapeutic application, it is very important to validate the safety of the method. CRSIPR/Cas9 system is thought to have off-target effects. Therefore, it is important to perform off-target analysis. In addition, TET3 could change the genome-wide methylation. In this study, off-target analysis, genome-wide methylation analysis, and genome-wide expression analysis were not performed. Only 7 genes were analyzed by RT-PCR. It is essential to perform genome-wide analysis (BS-seq, RNA-seq) and off-target analysis.

We agree with the reviewer, and we now performed genome-wide off-target analysis for both dCas9-TET3CD-*Rasal1*-sgRNA4 and the newly developed dHFCas9-TET3CD-*Rasal1*-sgRNA4 targeting constructs by CHIP-Seq analysis accordingly. This technique allowed for direct identification of 48 genes (including *Rasal1*) targeted by dCas9-TET3CD-*Rasal1*-sgRNA4 and of 8 genes (including *Rasal1*) targeted by dHFCas9-TET3CD-*Rasal1*-sgRNA4 (**Figure 2j, Table 1**). While CHIP-Seq has the advantage that results are not affected by secondary changes due to reactivation of *RASAL1* (as in RNA-Seq combined with BS-Seq), our results show that there is a dramatic advantage of using the newly developed high fidelity Cas9 which reduced off target effects by 85%, as described in the revised manuscript.

3. Although expression of Rasal1 is significantly induced in mouse model, improvement of fibrosis is very small. It suggests that genes other than Rasal1 could be involved in generation of the disease. In this point, it is important to examine whether Rasal1 knockout mouse show the fibrosis phenotype.

Thank you for this comment. In order to address the reviewer's concern we have now generated Rasal1 knockout mice by gene trapping strategy and obtained *Rasal1*^{tm1a/tm1a} "knockout first" mice, in which Rasal1 is decreased by 80% (**Figure 3a-c**). In these mice renal fibrosis was significantly increased as compared to wildtype mice upon challenge by UUO, providing the important causal link between lack of Rasal1 and kidney fibrosis (**Figure 3d**). As indicated above, however, we identified pro-fibrotic off target effects by our dCas9-TET3CD-*Rasal1*-sgRNA4 through a genome-wide search, which likely counteracted the therapeutic effect (**Table 1**). As described we therefore developed a novel endonuclease deactivated high-fidelity dHFCas9-fused to TET3 catalytic domain instead of dCas9. This reduced off target genes from 48 to 8, avoided targeting of pro-fibrotic genes and increased the therapeutic effect to an almost 50% reduction of fibrosis upon UUO challenge as compared to *LacZ* control construct (**Figure 5g-j**).

4. Although five sgRNAs were used for Rasal1, only sgRNA4 can demethylate the gene, suggesting that this method can be applied to only limited locus. To test whether this method can be generally applied to genes, it is important to perform systematically application of dCas9-TET3 in vivo is important.

In order to test utility of dCas9/dHFCas9-TET3 as general application, we now systematically tested a broad range of guiding RNAs for Rasal1, and found that, in general, the sense-location guided RNAs are more efficient than the anti-sense located guiding RNAs (**Figure 1c- 1e, 1i, Figure 2a-2b**). We next tested this approach in 3 other genes *in vitro* (human *EYA1*, human *LRFN2* and *Klotho* in both human mouse), which were previously identified as hypermethylated in kidney fibrosis, and successfully targeted all of these genes with a maximum of 8 guiding RNAs designed per gene (**Figure 1c-e, i**). In addition, dHFCas9-TET3CD-*Klotho*-sgRNA significantly improved kidney fibrosis *in vivo* (**Figure 6**). While the location for the best-suited guiding RNA is decided by multiple factors (such as chromatin structure, accessibility of the promoter region, etc.) we believe that we now show in the revised manuscript that our technology can be generally applied with reasonable effort both *in vitro* and *in vivo*.

5. The authors argue that TET3CD can demethylate longer range compared to TET1CD because TET3CD have larger domain than TET1CD. However, they did not show the range of demethylation using TET1CD in this locus. Because the range of demethylation could be different between loci, they should perform the experiments for several loci before generalize the discussion.

We agree with the reviewer that this hypothesis cannot be generalized based on our results. Because we believe that it is beyond the scope of this study to investigate different range of demethylation between TET1 and TET3, we have now removed this argument from the discussion.

6. The authors argue that TET3CD can demethylate longer range compared to TET1CD because TET3CD have larger domain than TET1CD. This line of investigation would be helpful to the scientific community. Unfortunately, there is no biochemical evidence for this. It would be better to show some biochemical evidence for this.

As outlined above, we believe that- even though we agree it would be helpful to the scientific community- it is beyond the scope of this study to address the question of how TET1 and TET3 differ with respect to range of demethylation.

Reviewer #2

This is technically well performed methods study that will be of interest to investigators wanting an approach for modifying DNA methylation in a sequence specific way.

Thank you.

However, it is not an entirely novel study since there are already recent publications reporting the use of CRISPR/Cas9 technology to modify methylation in a site specific manner, including demonstration that the technology works in vivo. Examples include Volta et al, NAR March 2016; Xu et al, Cell Discovery 2, 2016; and Shawn Liu et al in Cell Sept 2016. Hence there is a question as to whether the reporting of this technology is now sufficiently novel for the readership of Nature Communications.

There are now two main novelties of our revised study as compared to previous work:

1. Our study is the first to utilize gene-specific demethylation not only in vivo, but in a disease setting, and our data shows for the first time that a therapeutic effect is achieved by site-specific reversal of aberrant methylation.
2. All previous work in this respect (including our own study before revision) suffered from significant off-target effects of CRISPR/Cas9 inherent to the conventional Cas9 used in these studies. We have now for the first time used a modified high-fidelity Cas9 (dHFCas9) which was originally described by Kleinstiver et al¹, and we show that by using endonuclease deactivated dHFCas9, off-target effects were reduced by over 85%, and that the therapeutic effect using dHFCas9 was significantly improved as compared to conventional Cas9.

In addition, all previous techniques previously published are based on the use of the catalytic domain of TET1. We here for the first time show utility of TET3CD.

Also it is already established that RASAL1 is important in renal fibrosis and that its expression is controlled by DNA methylation, this information is nicely confirmed in the present study but again lacks novelty.

The focus of this study was not primarily to study the role of RASAL1 in renal fibrosis (even though the newly generated Rasal1 knockout mice presented in the revised manuscript now for the first time prove that lack of Rasal1 contributes to kidney fibrosis, **Figure 3**), but to establish gene-specific demethylation in a disease setting. We therefore purposefully utilized a system, where gene-specific hypermethylation (*Rasal1*) has been established to be relevant in kidney fibrosis^{2,3} to introduce gene-specific demethylation as novel therapeutic strategy.

We now also show utility of this approach using 3 additional genes (*EYA1*, *LRFN2* and *Klotho*) which were previously shown to be hypermethylated and relevant in kidney disease², and we applied the technology successfully in both human (*in vitro*, **Figure 1**) and mouse (both *in vitro* and *in vivo*, **Figures 2,5,6**).

The effects of targeted hydroxymethylation of RASAL1 were however modest and were only examined in one experimental model of fibrosis, this therefore limits the interest of the work to the wider fibrosis research community.

As outlined in detail above, through establishing a high-fidelity dHFCas9-TET3CD (thereby avoiding pro-fibrotic off-target effects) and through systematic analysis of different injection sites in the revised manuscript, we now substantially increased therapeutic effect to a 48.7% reduction of kidney fibrosis upon UUO as compared to *LacZ* control vector (**Figure 5**). While it has been shown that *Rasal1* methylation contributes to fibrosis also in the heart and liver⁴⁻⁶, aberrant promoter methylation of specific genes in general has been linked to a variety of diseases including leukemia, pancreatic and ovarian cancer, as well as neurodegenerative disease⁷⁻¹². In the revised manuscript we have now performed gene-specific demethylation and re-activation of 4 more genes, indicating that our technology can easily be applied to different genes. Thus, we believe that this study is of interest not only to the wider fibrosis research community but applicable to any diseases where aberrant gene methylation is causally involved.

The mechanisms by which hydroxymethylation altered RASAL1 expression were not fully examined and as such this work does not sufficiently advance our understanding of the molecular regulation of RASAL1 or in turn the mechanisms by which RASAL1 regulates fibrosis.

As stated above, we aimed to for the first time establish gene-specific demethylation in a disease setting, purposefully utilizing a system, where gene-specific hypermethylation (*Rasal1*) has been established², and where the mechanisms of how hydroxymethylation alters RASAL1 expression has already been identified³. We now tried to explain this better in the introduction of the revised manuscript. Nevertheless,

we are here the first to provide proof-of-principle that lack of Rasal1 increases kidney fibrosis by the newly generated Rasal1 knockout mice presented in the revised manuscript.

Reviewer #3

This is a very interesting manuscript about the design of a novel fusion protein using dead Cas9 fused to the catalytic domain of the TET3 enzyme to enable site specific promoter demethylation. The authors provide many controls to show specificity of their approach, replicate the approach in human and mouse, and then demonstrate proof of principle in the UUO model of kidney injury and fibrosis.

Overall, I think this is a fabulous Methods paper, demonstrating proof of principle and should be of appeal to the readership. The effect size on mouse kidney disease biology is very small, and of uncertain significance. Nevertheless, as a Methods paper, it will be of value to the community.

In addition, I have a number of significant concerns that should be addressable, to enhance the appeal and digestibility of the complicated biology.

Thank you for the positive assessment of our work. We hope you will agree that we undertook substantial effort to address all of your (and also the other reviewers') concerns as outlined in detail below.

1. It appears that the manuscript was prepared for Nature, with 3 figs and 18 supp figs. It is very hard to read in places, with poor explanation of terms, and concepts for a general reader, for whom Nat Comm is directed. For example bisulfite sequencing is poorly explained, hMeDIP is poorly explained, TK188 cells are poorly described, Fig 1B is poorly explained and defined. IN other places terms and explanations for approach are poorly explained to a non-specialist reader. I would propose that the text length is extended and some of the supp figures could be brought into the main text.

We have now extended the text and included more figures into the main text, and overall tried to better explain what we have done.

2. A diagram explaining the promoter, showing methylation and hydroxymethylation would be very helpful to the reader.

This is a good point, we have now done that (**Figure 1a**).

3. It would be helpful to demonstrate a lack of off target effect for the approach. So using specific guide4 and the fusion protein, can you show a lack of off target effects, at candidate off target sites in the genome?

Thank you, this is an extremely important point. We have now performed extensive

new experiments to address this concern (and also to reduce off target sites):

1. We have performed genome-wide off-target analysis by ChiP-Seq analysis of chromatin bound to our dCas9-TET3CD-*Rasa1*-sgRNA4 construct (using Myc-tag antibody pulldown) and indeed identified 47 genes in addition to *Rasa1* that were “off”- targeted by our construct. Importantly these genes also contained pro-fibrotic genes (**Table 1**), which may explain the relatively small anti-fibrotic effect observed using dCas9-TET3CD-*Rasa1*-sgRNA4.
2. We therefore utilized a novel high fidelity dCas9 (HF) to develop a dHFCas9-TET3CD-*Rasa1*-sgRNA4 construct (**Figure 2g**). Off-target analysis for this approach showed only 7 genes in addition to *Rasa1*, and anti-fibrotic effect was substantially increased by this new and improved approach (**Figure 2j, Figure 5, Table 1**). The use of high-fidelity Cas9 is a major novelty over previous work and appears to dramatically reduce off-target effects inherent to previously used CRISPR/Cas9 technology.

4. The text repeatedly states that RASAL1 methylation is causal in fibrotic disease, but since it is a DNA state triggered by disease and cytokine milieu, it could not be considered causal. It is in response to disease, not causal of disease. The use of this term should be modified, as it is misleading.

We agree and we have now modified the text accordingly. Also, in order to clarify causality of lack of *Rasa1* and fibrosis, we now additionally generated *Rasa1* knockout mice on expression level (*Rasa1*^{tm1a/tm1a}), which showed an 80% decrease in *Rasa1* expression (**Figure 3a-c**). Kidney fibrosis was significantly increased upon challenge with UUO as compared to *Rasa1*^{+/+} littermate control mice (**Figure 3d**), proving causality between lack of *Rasa1* and aggravation of fibrotic disease.

4. There is a large discrepancy between the effect of the approach shown in Fig 1E, 1L, and in 2G and 2H when compared to the apparent restoration of protein levels showing in Fig 2F. It would be more convincing to increase the number of replicates in Fig 2F so that the degree of protein restoration can be more clearly defined.

Done. We have now increased the number of replicates in these experiments (**Supplementary Figure 8d**)

5. Is there an explanation for why there is no endothelial or epithelial uptake of the viral reporter for sgRNA in Fig 3B. It seems unlikely that the virus does not get into other cell types and generate the fusion protein. Note DDK is not defined

This important question of what cell types take up the virus through which delivery method has now been systematically investigated in the revised manuscript:

For the revised manuscript we have established lentiviral vectors labelled with RFP

and GFP respectively and analysed viral uptake after 4 different delivery methods (into renal vein, into renal artery, into renal parenchyma, as well as retrograde into the ureter, **Figure 4**). Injection of lentivirus into the renal artery transduced the fewest cells overall among all four techniques in both healthy and UUO kidneys (**Figure 4c**). Venous injection lead to transduction of interstitial cells (**Figure 4e**), albeit to a lower extent as compared to intraparenchymal injection (**Figure 4d**). In contrast, retrograde injection into the ureter predominantly transduced tubular epithelial cells with high efficiency (**Figure 4f**).

These findings are in line with a study by Beronja et al. in mouse embryos which suggested that lentivirus does not pass the basal membrane¹³, and thus likely explain why in the previous version of the manuscript (where only injection into renal artery was performed) epithelial cells did not take up the virus.

6. The effect size of this approach on hard outcome measures, fibrosis, collagen I and α SMA is very small. I doubt that the findings are statistically significant at the $P < 0.0001$ level as claimed in Fig 3K-M. Given the limited utility of the UUO model, would it be worth considering a more chronic model to determine whether a larger effect can be achieved, such as the folate model the authors have used previously?

As outlined in detail above, the therapeutic effect could be dramatically increased through 1) the use of a new high fidelity Cas9 which reduced pro-fibrotic off-target effects imminent to the previously used dCas9 and 2) systematic optimization of lentiviral injection sites.

This lead to an almost 50% reduction of fibrosis, and even higher reduction of collagen I and α SMA in the UUO model (**Figure 5**).

We consciously did not use the folate model for our purpose because folate directly interferes with methylation in general and thus interference with the effects observed through our demethylation construct cannot be excluded.

References:

- 1 Kleinstiver, B. P. *et al.* High-fidelity CRISPR-Cas9 nucleases with no detectable genome-wide off-target effects. *Nature* **529**, 490-495, doi:10.1038/nature16526 (2016).
- 2 Bechtel, W. *et al.* Methylation determines fibroblast activation and fibrogenesis in the kidney. *Nat Med* **16**, 544-550, doi:10.1038/nm.2135 (2010).
- 3 Tampe, B. *et al.* Induction of Tet3-dependent Epigenetic Remodeling by Low-dose Hydralazine Attenuates Progression of Chronic Kidney Disease. *EBioMedicine* **2**, 19-36, doi:10.1016/j.ebiom.2014.11.005 (2015).
- 4 {Bian, B., E. B. *et al.* New advances of DNA methylation in liver fibrosis, with special emphasis on the crosstalk between microRNAs and DNA methylation machinery. *Cellular signalling* **25**, 1837-1844, doi:10.1016/j.cellsig.2013.05.017 (2013).

- 5 Xu, X. *et al.* Epigenetic balance of aberrant Rasal1 promoter methylation and hydroxymethylation regulates cardiac fibrosis. *Cardiovascular research* **105**, 279-291 (2015).
- 6 Tao, H. *et al.* MeCP2 controls the expression of RASAL1 in the hepatic fibrosis in rats. *Toxicology* **290**, 327-333, doi:10.1016/j.tox.2011.10.011 (2011).
- 7 {Tomar, T., T. *et al.* Genome-wide methylation profiling of ovarian cancer patient-derived xenografts treated with the demethylating agent decitabine identifies novel epigenetically regulated genes and pathways. *Genome medicine* **8**, 107, doi:10.1186/s13073-016-0361-5 (2016).
- 8 {Schoofs, S., T., Berdel, W. E. & Muller-Tidow, C. Origins of aberrant DNA methylation in acute myeloid leukemia. *Leukemia* **28**, 1-14, doi:10.1038/leu.2013.242 (2014).
- 9 {Guillamot, G., M., Cimmino, L. & Aifantis, I. The Impact of DNA Methylation in Hematopoietic Malignancies. *Trends in cancer* **2**, 70-83, doi:10.1016/j.trecan.2015.12.006 (2016).
- 10 Mishra, N. K. & Guda, C. Genome-wide DNA methylation analysis reveals molecular subtypes of pancreatic cancer. *Oncotarget* **8**, 28990-29012, doi:10.18632/oncotarget.15993 (2017).
- 11 Yao, B. *et al.* Epigenetic mechanisms in neurogenesis. *Nature reviews. Neuroscience* **17**, 537-549, doi:10.1038/nrn.2016.70 (2016).
- 12 {Kisiel, K., J. B. *et al.* New DNA Methylation Markers for Pancreatic Cancer: Discovery, Tissue Validation, and Pilot Testing in Pancreatic Juice. *Clinical cancer research : an official journal of the American Association for Cancer Research* **21**, 4473-4481, doi:10.1158/1078-0432.CCR-14-2469 (2015).
- 13 Beronja, S., Livshits, G., Williams, S. & Fuchs, E. Rapid functional dissection of genetic networks via tissue-specific transduction and RNAi in mouse embryos. *Nat Med* **16**, 821-827, doi:10.1038/nm.2167 (2010).

Reviewers' comments:

Reviewer #1 (Remarks to the Author):

Comments to the authors

NCOMMS-16-22492A-Z

Gene-specific hydroxymethylation via CRISPR/Cas9-TET3 rescues expression of methylated RASAL1 and attenuates experimental fibrosis.

The manuscript was greatly improved and suitable for publication in Nature Communications.

Reviewer #4 (Remarks to the Author):

Interesting new technology, but very little in terms of new biology or understanding of fibrosis.

Reviewer #5 (Remarks to the Author):

This is an interesting a novel study demonstrating the therapeutic potential of a single construct delivered by lentiviral transduction to achieve gene specific promoter demethylation using a chimeric dHFCas9-TETCD enzyme with sgRNA expression. Much of the work centres on the development of the construct and in validating on and off target effects.

The issues raised in the previous review have been, in the most part, well addressed. The new data considerably strengthen the study. However, I have a couple of points to be addressed.

1. The parenchymal injections of lentivirus were given at the time of UUO surgery when resident fibroblasts are few in number. Is it known from the CMV-GFP study what proportion of the large number of fibroblasts/myofibroblasts accumulating on day 10 UUO expressed GFP (or expressed Cas9 from transduction with the dHFCas-9-TETCD constructs)?

2. Please provide clarification on the statistical methods used in the analysis of the fibrosis endpoints for the UUO models in Figures 3, 5 and 6. In particular, Fig 3d shows data for mean +/- SD for Masson trichrome, collagen I and a-SMA staining. These error bars are overlapping and only groups of 6 mice were used. It is difficult to see how a significance of $P < 0.0001$ can be claimed between WT UUO and RASAL1 KO UUO if using one way ANOVA with Bonferroni's multiple comparison test (as indicated in the methods). Similarly, the differences in collagen I and a-SMA are very minor in Fig 6g and 6h – are they actually different?

Reviewer #1:

The manuscript was greatly improved and suitable for publication in Nature Communications.

Thank you for this positive assessment of our work and effort.

Reviewer # 4:

This is a study that explores the role of high-fidelity CRISPR/Cas9-based gene-specific hydroxymethylation to activate endogenous anti-fibrotic genes in a UUO model of kidney fibrosis.

Thank you, this is correct.

One limitation of the study is that the genes of interest are already well known regulators of kidney fibrosis, and the main advance is in the gene editing technology to de-repress these genes as a potential therapeutic approach.

The focus of this study was to provide proof of principle for our novel gene editing technology in a well-established and well-studied disease model. We therefore chose genes encoding for well-known regulators of kidney fibrosis, which had been previously shown to be hypermethylated within fibrotic kidney disease, for our technology. The technology itself, however, can of course be applied to any disease where aberrant gene methylation plays a role.

1. Abstract "...providing the first gene-specific demethylating technology in a disease model." is an overstatement. CRISPR/dCas9-TET for in vivo DNA demethylation has been reported (Morita S et al, Nat Biotechnol. 2016, ref#45). Previous studies have demonstrated the critical role of RASAL1 in animal model of kidney fibrosis. The significance of this study lies in a new approach for the control of RASAL1 and Klotho gene expression.

In the referenced manuscript by Morita S et al., the authors provide proof of principle of gene editing *in vivo* by introducing an editing vector into the brain of mouse fetuses by electroporation. They did, however, not use a disease model. Nevertheless, we now adjusted the abstract according to this comment.

2. Introduction and results: The authors have previously shown that overexpression of TET3 reactivated various genes, including RASAL1, in diseased kidney and attenuated experimental kidney fibrosis (refs 5,14). However, as a control in this study, the authors demonstrated that overexpression of TET3CD had no significant gene induction in any of 4 genes, including RASAL1. The authors concluded that demethylation is not due to overexpression of TET3CD (Supplementary Fig.3, text at the bottom of page 5). More confusingly, Fig.2i showed that both dCas9-TET3CD and

dHFCas9-TET3CD attenuated promoter methylation and rescued Rasal1 and Kl gene expression. These data seemed to be contradictory one another and need to be clarified.

It is important to note that unlike in previous studies (where we have indeed shown that overexpression of full TET3 reactivates various genes), in this study we only used the catalytic domain of TET3 (TET3CD), without the DNA binding domain. Overexpression of TET3CD alone (without the DNA binding domain and without guiding RNA) has no effect (as shown in Supplementary Figure 3). In Figure 2i TET3CD is fused to guiding RNAs which then allows demethylation of specific genes. We have now modified the labeling within the figures 1g, 1j-k and 2c, 2e and 2h-l accordingly to make these important distinctions clearer.

3. Results: The purpose of studying Rasal1^{tm1a/tm1a} mice (Fig. 3) was not clear.

We generated and studied Rasal1^{tm1a/tm1a} mice as per the request of reviewer #1, who rightfully argued that loss of RASAL1 had not yet been causally linked to progression of kidney fibrosis. Our mice now show for the first time that indeed loss of Rasal1 leads to increased kidney fibrosis upon challenge with UUO.

4. Discussion: The essential role of glycosylase enzymes in active DNA demethylation was not discussed. In the process of active DNA demethylation, TET enzymes mediate iterative oxidation of 5mC to generate 5fC and 5caC, which can be excised by TDG to generate abasic sites as part of the DNA repair pathway that ultimately regenerates unmodified C. 5fC and 5caC are excised by glycosylase enzymes, such as thymine DNA glycosylase (TDG), to generate an abasic site. Cells then use the base excision repair (BER) mechanism to regenerate unmodified C. The findings that simultaneous overexpression of TET and TDG in the HEK293 cell line depleted TET-associated 5caC and 5fC (He et al. Science 2011; Nabel et al. Nature Chem. Biol. 2012) demonstrated a requirement for TDG in a complete pathway for active DNA demethylation.

We agree that the mechanism of how glycosylase enzymes (and especially TDG) are involved in active TET3-induced DNA demethylation is important. This has been addressed by us in a previous study (Tet3 induces formation of 5-formyl-cytosine and 5-carboxyl-cytosine, subsequent base excision by Tdg, and finally replacement with unmodified cytosine, see Tampe B et al. EBioMedicine 2015), which is referenced in the manuscript. We now discuss this aspect in more detail in the revised manuscript.

Reviewer #5:

This is an interesting and novel study demonstrating the therapeutic potential of a single construct delivered by lentiviral transduction to achieve gene specific promoter demethylation using a chimeric dHFCas9-TETCD enzyme with sgRNA expression. Much of the work centres on the development of the construct and in validating on and off target effects.

The issues raised in the previous review have been, in the most part, well addressed. The new data considerably strengthen the study.

Thank you.

However, I have a couple of points to be addressed.

1. The parenchymal injections of lentivirus were given at the time of UUO surgery when resident fibroblasts are few in number. Is it known from the CMV-GFP study what proportion of the large number of fibroblasts/myofibroblasts accumulating on day 10 UUO expressed GFP (or expressed Cas9 from transduction with the dHFCas-9-TETCD constructs)?

Thank you for this insightful comment. In response, we have now performed double immunolabeling of parenchymal CMV-GFP injected kidneys 10 days after UUO surgery for GFP and α SMA, and find that an average of 46% of all α SMA-positive cells are also positive for GFP. This information is now included as new Figure 4g in the revised manuscript.

2. Please provide clarification on the statistical methods used in the analysis of the fibrosis endpoints for the UUO models in Figures 3, 5 and 6. In particular, Fig 3d shows data for mean \pm SD for Masson trichrome, collagen I and α -SMA staining. These error bars are overlapping and only groups of 6 mice were used. It is difficult to see how a significance of $P < 0.0001$ can be claimed between WT UUO and RASAL1 KO UUO if using one way ANOVA with Bonferroni's multiple comparison test (as indicated in the methods). Similarly, the differences in collagen I and α -SMA are very minor in Fig 6g and 6h – are they actually different?

We are thankful for this important comment and apologize for the confusion. Throughout the manuscript, histological analysis was assessed as 10 visual fields per mouse and average values of each mouse were calculated. The average values for each mouse (n=6) was then used to calculate average per group. For Figure 3, however, in the previous version of the manuscript, the total number of visual fields (n=10) per mouse (n=6) went into the statistical analysis (60 values total). In the revised manuscript we have now performed statistical analysis in accordance with all other Figures in the manuscript. Statistical analysis was performed using One-way ANOVA with Bonferroni post-hoc analysis comparing selected pairs of columns. We now added this important information to the revised manuscript.

Also, we again re-evaluated statistical analysis for collagen I and α -SMA in Fig.6g and 6h (where Klotho methylation was targeted), and find that while differences are smaller than when Rasal1 methylation was targeted (Fig. 5i-j) these are indeed significant.

REVIEWERS' COMMENTS:

Reviewer #4 (Remarks to the Author):

The authors have satisfactorily addressed my concerns.

Reviewer #5 (Remarks to the Author):

The issues raised have been satisfactorily addressed. The demonstration that just under half of the myofibroblasts were transduced on day 10 UUO gives greater weight to the degree of reduction of renal fibrosis seen with the dHFCas-9-TETCD construct.

Reviewer #4 (Remarks to the Author):

The authors have satisfactorily addressed my concerns.

Thank you.

Reviewer #5 (Remarks to the Author):

The issues raised have been satisfactorily addressed. The demonstration that just under half of the myofibroblasts were transduced on day 10 UUO gives greater weight to the degree of reduction of renal fibrosis seen with the dHFCas-9-TETCD construct.

Thank you for the positive evaluation of our revised manuscript and our effort.